# Evolution of core archetypal phenotypes in progressive high grade serous ovarian cancer

Aritro Nath [1], Patrick A. Cosgrove [1], Hoda Mirsafian[1], Elizabeth L. Christie[2,3], Lance Pflieger [1], Benjamin Copeland[1], Sumana Majumdar[1], Mihaela C. Cristea[1], Ernest S. Han[4], Stephen J. Lee[4], Edward W. Wang[1], Sian Fereday [2,3], Nadia Traficante[2,3], Ravi Salgia[1], Theresa Werner[5], Adam L. Cohen[5], Philip Moos[6], Jeffrey T. Chang [7], David D. L. Bowtell[2,3✉] & Andrea H. Bild [1✉]

The evolution of resistance in high-grade serous ovarian cancer (HGSOC) cells following chemotherapy is only partially understood. To understand the selection of factors driving heterogeneity before and through adaptation to treatment, we profile single-cell RNA-sequencing (scRNA-seq) transcriptomes of HGSOC tumors collected longitudinally during therapy. We analyze scRNA-seq data from two independent patient cohorts to reveal that HGSOC is driven by three archetypal phenotypes, defined as oncogenic states that describe the majority of the transcriptome variation. Using a multi-task learning approach to identify the biological tasks of each archetype, we identify metabolism and proliferation, cellular defense response, and DNA repair signaling as consistent cell states found across patients. Our analysis demonstrates a shift in favor of the metabolism and proliferation archetype versus cellular defense response archetype in cancer cells that received multiple lines of treatment. While archetypes are not consistently associated with specific whole-genome driver mutations, they are closely associated with subclonal populations at the single-cell level, indicating that subclones within a tumor often specialize in unique biological tasks. Our study reveals the core archetypes found in progressive HGSOC and shows consistent enrichment of subclones with the metabolism and proliferation archetype as resistance is acquired to multiple lines of therapy.

[1] Department of Medical Oncology and Therapeutics, City of Hope Comprehensive Cancer Center, Monrovia, CA, USA. [2] Peter MacCallum Cancer Centre, Melbourne, VIC, Australia. [3] Sir Peter MacCallum Department of Oncology, The University of Melbourne, Melbourne, VIC, Australia. [4] Division of Gynecologic Oncology, Department of Surgery, City of Hope, Duarte, CA, USA. [5] Division of Oncology, Department of Medicine, Huntsman Cancer Institute, University of Utah, Salt Lake City, UT, USA. [6] Department of Pharmacology and Toxicology, University of Utah, Salt Lake City, UT, USA. [7] Department of Integrative Biology and Pharmacology, University of Texas Health Science Center at Houston, Houston, TX, USA. ✉email: david.bowtell@petermac.org; abild@coh.org

Transcriptional dysregulation is a hallmark feature and a driver of evolution in human cancers[1]. As one of the deadliest forms of gynecological malignancy, the survival rates for high-grade serous ovarian cancer (HGSOC) have remained poor over the past few decades[2]. Despite initial responsiveness to platinum-based chemotherapy and the introduction of novel combination therapeutic interventions[3], the development of resistance over the course of treatment remains a major challenge in the clinical management of HGSOC[4,5]. Thus, characterizing the key transcriptional changes in HGSOC tumor evolution is critical for understanding tumor progression and resistance to cancer therapy[6,7].

A majority of HGSOCs arise from the epithelium of fallopian tubes[8], often resulting in the detection of malignant cells that escape into the fluids accumulating in the peritoneal cavity (ascites) or the lung pleural effusions following late-stage extra-abdominal metastases[9]. The genome of HGSOCs is characterized by somatic alterations leading to the loss of function of the tumor suppressor gene *TP53*[10,11] and regulator components of homologous recombination (HR) DNA-damage repair pathway, including *BRCA1* and *BRCA2*[12]. Whole-genome sequencing (WGS) analyses have revealed several key genomic mechanisms of acquired resistance, such as somatic alterations in the multi-drug resistance gene *ABCB1*, secondary somatic mutation alterations in HR genes, and the protection of stalled replication forks[13–15]. However, known mechanisms explain only a fraction of resistance drivers[13]. Therefore, focusing on transcriptional changes could help improve our understanding of chemoresistance, especially in cases where obvious single-gene alterations are not detectable. Further, the number of critical signaling pathways important for HGSOC cell growth and survival is unknown; therefore, therapeutic regimens may miss important oncogenic traits and enable progression.

Rapid developments in single-cell RNA-sequencing (scRNA-seq) technologies have enabled the investigation of intratumor heterogeneity and evolution at the cellular level[16–18]. Longitudinal analysis of tumors in response to drug treatment using scRNA-seq combined with DNA sequencing has been utilized to understand the ecology and evolution of tumors along with phenotypic mechanisms that could be harnessed as potential drug targets in resistant tumors[19]. Key questions in heterogeneous HGSOC that remain to be addressed include: (1) the number of key phenotypic features in progressive tumors, (2) the biological processes underlying progression, (3) how changes in the number of cells specializing in specific phenotypes contribute to progression, and (4) how genetically distinct subclonal populations impact phenotypic diversity. Until recently, a computation framework to identify these factors was insufficient.

Recent developments in cancer evolutionary theory suggest that tumor cells can evolve to exhibit a range of phenotypes under selective pressure such as chemotherapy[20,21]. However, each cell in the tumor may exist in a limited range of transcriptional states optimal at performing phenotypic tasks critical to survival owing to metabolic and spatial constraints[21]. Thus, elucidating the biological tasks associated with transcriptional cell specialists in chemoresistant HGSOC could help in developing new therapeutic strategies targeting these emergent phenotypes. To identify the number and biological function of tasks associated with HGSOC cancer cell progression, we employed a method that uses the Pareto optimization concept, which states that, when a combination of tasks dominates an organism's fitness, but the organism cannot be optimal at all tasks at once due to trade-offs, optimal phenotypes should fall on low-dimensional shapes called polyhedra. The number of vertices reflects the number of tasks essential to the fitness of the organism[21]. The approach defines a polytope where the number of vertices reflects the number of

tasks describing the data. The cells at the edges of the polytope (termed archetype) specialize in a specific biological task[22]. Based on these principles, we project the scRNA-seq profiles from the HGSOC samples on to archetypes to determine the number of driver phenotypes in the data, the biological features of those archetypical phenotypes, and to study if cells specialize in specific archetypical tasks during progression. Finally, these archetypes are evaluated together with genetic alterations to identify the potential link between somatic alterations and phenotypic state.

In this study, we use malignant ascites and pleural effusion samples from nine HGSOC patients, collected over months to years of treatment, to perform scRNA-seq and WGS analysis. We also perform scRNA-seq analysis of an independent cohort of unmatched eight pre-treatment and six post-treatment samples to study longitudinal patterns of transcriptomic heterogeneity in treatment naive compared to heavily treated tumors. Our results show enrichment of an archetype associated with elevated metabolic activity, driven by oxidative phosphorylation or glycolysis, and proliferation in post-treatment patients compared to treatment-naive patients of the validation cohort. Further, cellular defense response (CDR) and DNA repair describe two additional key archetypical phenotypes in HGSOC. While consistent genomic alterations do not define the archetypes, subclonal clusters inferred from scRNA-seq profiles are associated with the enrichment of the metabolic archetype as cancer cells progress on therapy. Finally, we validate the metabolic archetype activity in progressive HGSOC patient tumor samples using in vitro metabolic assays.

## Results

**Temporal transcriptomic diversity of HGSOC cells**. To study the landscape of genetic and transcriptomic heterogeneity of ovarian cancer cells in response to chemotherapy, we obtained 25 malignant ascites or pleural effusion samples from nine HGSOC patients (Fig. 1a and Supplementary Data 1–3). Samples were collected over the course of treatment, with initial samples obtained at the time of surgery or before the commencement of therapy in five of the nine patients, and early in treatment for the remainder. Following initial debulking surgery, patients received adjuvant platinum- and taxane-based chemotherapy as a first-line treatment followed by three to seven lines of chemotherapy over the course of their disease progression (Fig. 1a and Supplementary Data 1–3). The samples were processed to isolate nuclei or whole cells to perform scRNA-seq, WGS, and establish in vitro cell lines for metabolic assays (Fig. 1b and "Methods"). In addition, we obtained eight pre-treatment and six post-treatment malignant ascites or pleural samples as a validation cohort for the scRNA-seq analyses (Supplementary Data 4).

We analyzed the transcriptomes of ~36K high-quality cells or nuclei using scRNA-seq. Preliminary clustering of the scRNA-seq data resulted in the separation of cells by patients (Supplementary Figure 1). Following batch correction with canonical correlation analysis (CCA)[23], unsupervised clustering resulted in eight clusters representing individual cell types instead of patient identity (Fig. 1c). Reference-based prediction of cell types[24] revealed large clusters of predominantly malignant epithelial cells (clusters 0–2 and 7), confirmed by the expression of the epithelial marker (*EPCAM*) and the tumor biomarker *MUC16* (CA-125) (Supplementary Figure 2). Despite prior immune depletion, we also detected smaller but distinct clusters of immune cells, including monocytes and macrophages (cluster 3), CD4[+]/CD8[+] T cells (cluster 4), fibroblasts (cluster 5) and natural killer (NK) cells (cluster 6). In total, ~27K cells were identified as malignant epithelial cells that were distributed across multiple clusters reflecting the heterogeneity within the population

**Fig. 1 Longitudinal cohort sample collection and scRNA-seq analysis. a** Timeline of patients included in the study. Gray timeline shows days ×100, with 0 referring to the day primary surgery was performed. The red lines indicate CA-125 levels, while the triangles along the timelines indicate the time points at which malignant fluid samples were obtained. Colored bars below the patient timeline indicates drug treatment received. **b** The malignant fluid samples were processed to remove immune and apoptotic cells, and processed for whole-genome sequencing, in vitro metabolic assays, and scRNA-seq. A portion of this figure panel was created using BioRender.com. **c** Uniform Manifold Approximation and Projection (UMAP) and clustering of the integrated high-quality cells profiled using iCell8 (patients 1–3) or 10X (patients 4–9) scRNA-seq platforms following CCA normalization. The numbers indicate clusters obtained following unsupervised clustering. Stacked bar plots on the right show the distribution of various predicted cell types across clusters and the diversity of samples distributed within each cluster. The colors on the UMAP indicate the cell-type classification of the single cells.

(Fig. 1c). We analyzed patterns of expression changes over time and found that few differentially expressed genes were commonly shared across patients (Supplementary Figure 3). This pattern could also reflect the sparsity of scRNA-seq data that contributed to the observed lack of consistent changes across patients in the high-dimensional gene expression space. Therefore, we next adopted an approach to project the scRNA-seq data in a low-dimensional space and investigated the evolution of key phenotypes.

**Transcriptional evolution of ovarian cancer cells is associated with core biological tasks.** To interrogate transcriptional heterogeneity in progressive HGSOC, we applied an approach that accounts for tumor evolution and the use of tasks to enhance fitness[21]. Our goal was to determine how many archetypal phenotypes are found in HGSOC and how these tasks evolve as patients receive therapy and become resistant. We utilized a Pareto task inference method that relies on the principle convex hull algorithm to identify core archetypes[25]. Briefly, the method attempts to identify a polytope that can best enclose the principal component projection of the gene expression data. The vertices of this polytope are inferred as the core archetypes. Analyses were limited to scRNA-seq profiles obtained using the 10X platform (patients 4–9) with sufficient numbers of malignant epithelial cells available for projection on to the archetypes. To determine the shape of a polytope that can best enclose the data, we fitted polytopes with a varying number of vertices ranging from 3 to 8 (Supplementary Figure 4). There was a minimal gain in the variance explained by the models with >3 archetypes, showing that a triangle was enough to enclose the data. Moreover, any gain in variance explained by the models with >3 archetypes was at the

cost of increased uncertainty in the position of vertices, and a decrease in the ratio of the volume of the polytope to the convex hull (t-ratio) confirmed that the three-vertex triangle reliably enclosed the data (Fig. 2a).

To determine the distinct biological tasks associated with each archetype, we implemented a multitask learning approach based on group-lasso (see "Methods" for details) that applied the hallmark pathways and genes to cells located on each archetypal vertex. Hierarchical clustering analyses with pathway coefficients show three distinct clusters linked to the archetypes (Fig. 2b and Supplementary Figures 4 and 5). Three broad tasks were associated with these archetypes, including metabolism and proliferation (MAP), cellular defense response (CDR), and DNA-damage repair (DDR) (Fig. 2c). The MAP archetype was defined based on the enrichment of multiple key metabolism phenotypes, including glycolysis, oxidative phosphorylation, and proliferation pathways associated with cell cycle and E2F genes, G2M checkpoint genes, epithelial-to-mesenchymal transition (EMT), and MYC targets (Supplementary Figures 5 and 6). Association of key genes indicative of proliferation (*MKI67*) and glycolysis (*GAPDH*)[26] supported the classification of this archetype. The CDR archetype was defined based on enrichment of the interferon-γ response pathway[27] and the enrichment of multiple downstream pathways and genes related to the activation of CDR, including canonical IL-6/JAK/STAT3[28] and interferon-γ signaling pathways, as well as cell cytokine and immunogenic signaling genes. Finally, the DDR archetype was derived based on enrichment of apoptosis, P53, and TNF-alpha related[29] signaling pathways along with key genes such as *ATM*[30] and *CHEK1*[31]. The classification of the archetypes using the lasso approach shrinks coefficients of several pathways to zero; therefore, we also evaluated the contribution of

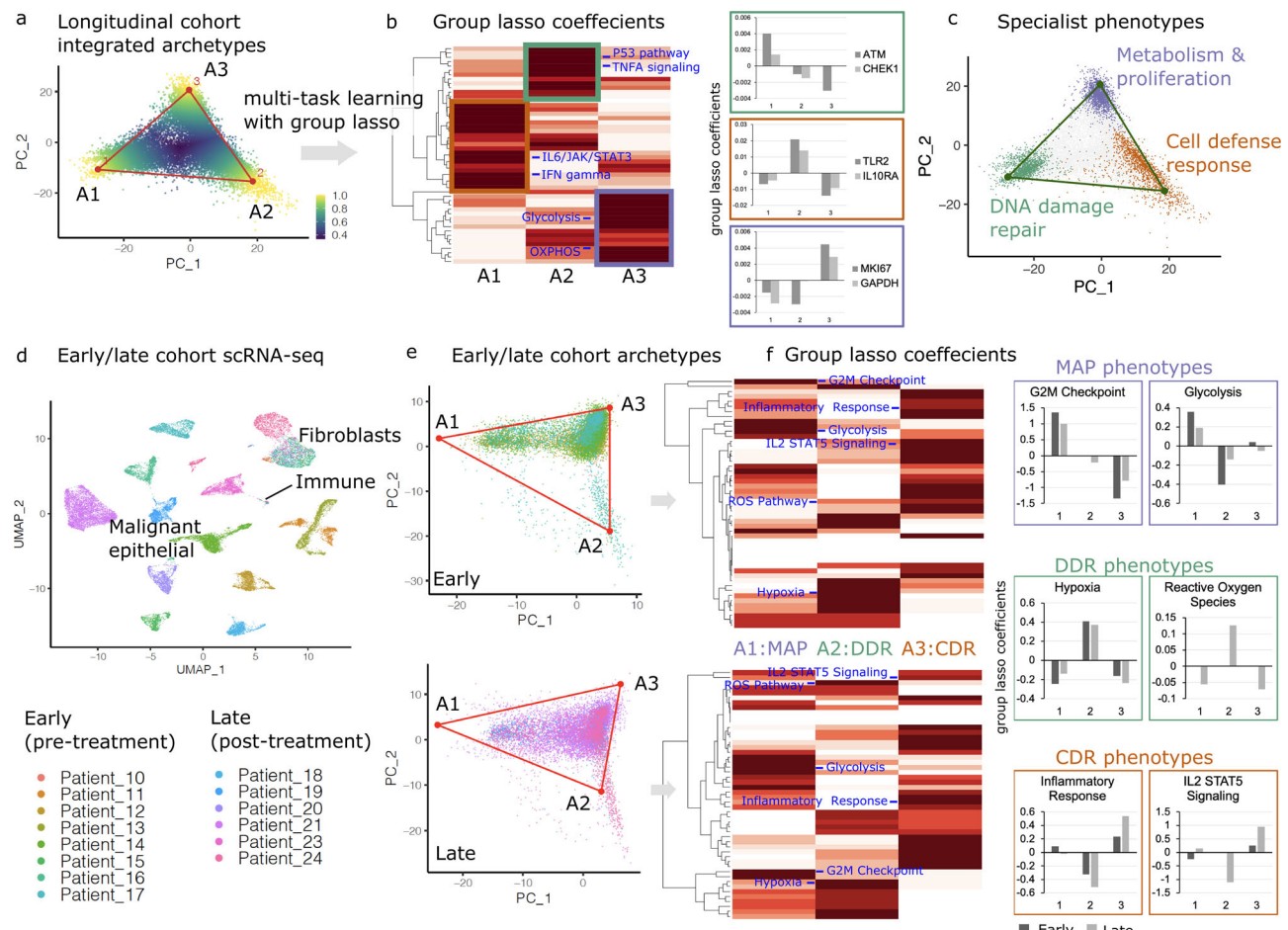

**Fig. 2 Archetype analysis to investigate shifts in biological tasks over time. a** Cells from the integrated longitudinal cohort are projected on the first two principal components of scRNA-seq data. Each vertex identified using the Pareto task inference algorithm represents an archetype specializing in a specific biological task. Single cells are colored by maximum archetype score (standardized and inversed Euclidean distance to the vertex), revealing proximity to the nearest archetype. **b** Heatmap showing the hierarchical clustering of group-lasso coefficients for various hallmark pathways associated with each archetype. The three colored boxes display three hierarchical clusters of related phenotypes that reveal the identity of the biological task associated with each archetype. The principal pathways that define the archetypes are indicated on the heatmap in blue, with the green box indicating DNA-damage repair (DDR), the red box indicating cellular defense response (CDR), and the blue box indicating metabolism and proliferation (MAP) archetypes. The inset bar plots show the group-lasso coefficient (Y-axis) of key genes corresponding to the three archetypes (X-axis) from the scRNA-seq data of the longitudinal cohort patients. Source data for single gene coefficients are provided as Source Data file. **c** Three major biologicals tasks were predicted using the multitask learning approach. Specialist cells are colored according to the closest archetype (cells in the 95th percentile), whereas non-specialists distal from all three archetypes are shown in gray. **d** Dimensionality reduction (UMAP) projections and clustering of the integrated 28K high-quality cells, including 21K malignant epithelial cells, from the early/late (pre-/post-treatment) cohort profiled using 10× platforms. Patients 10–17 were treatment-naive, while patients 18–24 received multiple lines of treatment. Major predicted cell types are annotated to show separate clusters with malignant epithelial cells, immune cells, and fibroblasts. **e** Archetypes are annotated on the principal component projections of the early and late cohorts. The samples from each cohort are shown separately to clearly display the population shifts in the archetypes. **f** Hierarchical clustering of group-lasso coefficients for archetypes determined for early and late cohort patients, with principal pathways defining the archetypes indicated in blue. The inset bar plots on the right show group-lasso coefficients (Y-axis) of key pathways that were used to define the phenotypes associated with each of the three archetypes (X-axis).

individual signaling pathways to each archetype using regression analysis. We analyzed the association between KEGG pathway enrichment scores in single cells against the archetype scores of all three archetypes across all cells are listed in Supplementary Data 5. This second orthogonal approach to analyze the pathways uncovered consistent results. The classification of the MAP archetype was supported by the positive associations (positive coefficients and false discovery rate (FDR) < 0.05) with enrichment of key signaling pathways, including cell cycle, DNA replication, glycolysis, and oxidative phoshporylation. The CDR archetype was supported by positive associations with key immune response pathways, including NK cell-mediated

cytotoxicity, TLR receptor signaling, RIG-I (retinoic acid-inducible gene I) like receptor signaling, NOD-I (nucleotide-binding oligomerization I)-like receptor signaling, and T/B cell receptor signaling. Interestingly, the CDR archetype scores were also positive correlated with metabolism pathways of glycolysis and oxidative phosphorylation, but not with proliferation as indicated by cell cycle and DNA replication, indicating a decoupling of these two phenotypes in the CDR cells and also supporting the separate classification of this cell state from MAP. Furthermore, similar to the classification of the PI3K/MTOR and WNT pathways in the CDR archetype in the group-lasso analysis, the CDR cells were again associated with the KEGG MAPK and

WNT signaling pathways, suggesting that activation of these pathways contributed to the CDR archetype over MAP.

Across the single cells, the key pathways contributing to the MAP archetype (Supplementary Figure 7A) were positively intercorrelated (Pearson's correlation coefficient > 0, FDR < 0.05) (Supplementary Figure 7B), with subtle differences observed between subpopulations of cells classified based on cell cycle states (Supplementary Figure 7C) and between patients across time (Supplementary Figure 7D, E). For example, the MAP cells were more metabolically active in S and G1 cells compared to G2M cells, as expected. The MAP phenotypes on average were largely consistent over time, with some significant shifts observed within specific patients, like decreased glycolysis in patient 5, and increased glycolysis and oxidative phosphorylation in patient 8. The key pathways contributing to the CDR pathways were also highly intercorrelated (Pearson's correlation coefficient > 0, FDR < 0.05) (Supplementary Figure 8A), with some differences observed in specific patients over time (Supplementary Figure 8B). In particular, patients 5 and 9 showed reduced enrichment of multiple immune response pathways over time.

We also performed archetype analysis on an independent validation cohort of eight unmatched pre-treatment and six post-treatment malignant ascites or pleural effusion samples. All pre-treatment samples were from treatment-naive patients. Post-treatment patients received an average of five lines of treatment, including chemotherapies and targeted therapies (Supplemental Table 4). Dimensionality reduction and clustering of the cells from the early (pre-treatment)/late (post-treatment) cohort resulted in a large malignant cell cluster of epithelial origin and smaller immune cell and fibroblast clusters (Fig. 2d), confirmed by expression of individual markers (Supplementary Figure 9). As with the initial longitudinal cohort, we performed the Pareto task inference analysis on the validation cohort samples revealing three major archetypes in both the early and late cohorts (Fig. 2e and Supplementary Figure 10). Based on the key phenotypes specifically enriched in each archetype, we confirmed that the archetypes in the validation cohort also corresponded to MAP, DDR, and CDR (Fig. 2f and Supplementary Figure 11).

Next, we confirmed the presence of the three archetypes detected in the integrated scRNA-seq data in individual patients by resolving the scRNA-seq profiles of each patient from the longitudinal cohort. Comparing polytopes with a range of vertices confirmed a three-vertex polytope was once again sufficient to enclose the complete data for each patient (Supplementary Figure 12). The biological phenotypes associated with these three clusters were consistent with the phenotypes associated with the archetypes identified in the integrated dataset when performed separately for each patient (Fig. 3a, b and Supplementary Figure 13). Importantly, these analyses show that the three archetypes are a common feature of all HGSOC tumors, albeit with varying proportions over time.

We then evaluated the patterns of shifts in the populations of specialist cells, defined as cells close to a vertex representing one of the key archetypes (MAP, CDR, or DDR) during treatment of our initial patient cohort (Fig. 3c and Supplementary Data 6). At most time points, cancer cells were present that specialized in each of the three key archetypes, with three of five patients having an enrichment in either the MAP or CDR specialists. In the case of patient 4, all three archetypes were present at the three time points. MAP was the principal archetype at the first time point (42%), and with most cells specializing in the CDR (35%) or MAP (24%) archetypes at the last time point. The proportions of MAP archetype were higher at time 1 and 3 compared to time 2, which coincides with the lower CA-125 levels of the patient while on treatment during time 2 (Fig. 1a). The relative proportion of the specialists in patient 5 did not change over time, with MAP

remaining the critical archetype at the last time point (46%). Patient 6 also showed a pattern of MAP archetype enrichment that shows similar trends as CA-125 burden, with the highest levels at time 1 (30%) and time 3 (32%) compared to time 2 (11%). In the case of patient 7, CDR emerged as the core archetype at the second time point (81%). Although the two samples for patient 7 were collected across a gap of >3 years, the CA-125 levels were relatively low compared to the late time points of other patients (Supplementary Data 3), thus explaining the exceptional pattern of MAP specialists observed in this patient. The archetypes of patient 8 were mostly dominated by non-specialist cells (78%) at the last time point. However, this patient also showed an increase in the proportion of MAP specialists from 2 to 5% between the first and last time point, with the highest proportion of MAP specialists (16%) coinciding with the highest CA-125 levels for this patient at time 2 (Fig. 1a and Supplementary Data 3). In the case of patient 9, the proportion of MAP specialists increased progressively over time from 9 to 22%, again following the CA-125 levels for this patient. Thus, the patterns of the shift in the MAP archetype derived from the scRNA-seq data agree well with the levels of tumor marker over time (Supplementary Figure 14).

We also compared the overall distribution of the archetype specialists across single-cell transcriptional clusters of malignant cells and their distribution over time (Supplementary Figure 15A,B). Each specialist population was associated with a distinct transcriptional cluster, from six total clusters (Supplementary Figure 15A, lower left UMAP). The CDR specialists were largely associated with malignant cell cluster 0, while DDR specialists were associated with cluster 1 and MAP specialists with cluster 2 (Supplementary Figure 15B, left barplot). Clusters 3, 4 and 5 were small clusters with few cells, which were present transiently at time 1 or time 2 (Supplementary Figure 15B, right barplot). At the first time point, cluster 2 and MAP specialists were present at the lowest proportions compared to other archetypes, indicating that the cluster 2/MAP specialists were acquired along the course of progression (Supplementary Figure 15B, right barplot).

The proportion of specialists changed over time in patients. Specifically, the proportion of the MAP and CDR archetypes showed a strong negative correlation to each other (Pearson's correlation = −0.95, $R^2 = 0.9$) (Supplementary Figure 15C). This relative shift between the archetypes was correlated with the overall survival of the patients (time to death in days since primary surgery was performed). We found a negative correlation between the proportion of MAP specialists in a patient at the final time point and overall survival, with more MAP specialists associated with worse outcome (Supplementary Figure 15D). Although this correlation was not statistically significant in this small sample ($P = 0.1$), the observed effect size (Pearson's correlation = −0.69, $R^2 = 0.48$) calls for validation in a sample of larger size to assess clinical significance.

**The metabolic and proliferative archetype is enriched in late-stage resistant HGSOC cancer compared to treatment-naive cancer cells.** We next compared the specialist populations in the unpaired treatment-naive and post-treatment validation cohorts. CDR specialists were most common in all eight treatment-naive patients (average 33%), while MAP specialists were least common in seven out of eight treatment-naive patients (average 9%) (Fig. 4a and Supplementary Data 6). In contrast, MAP specialists were the principal archetype in three out of six post-treatment patients (average 22%). This reflected a significant shift towards the MAP archetype in post-treatment samples ($P = 0.008$), while CDR archetype decreased significantly in the post-treatment samples ($P = 1.5 \times 10^{-6}$) (Fig. 4b). The dramatic decrease in CDR

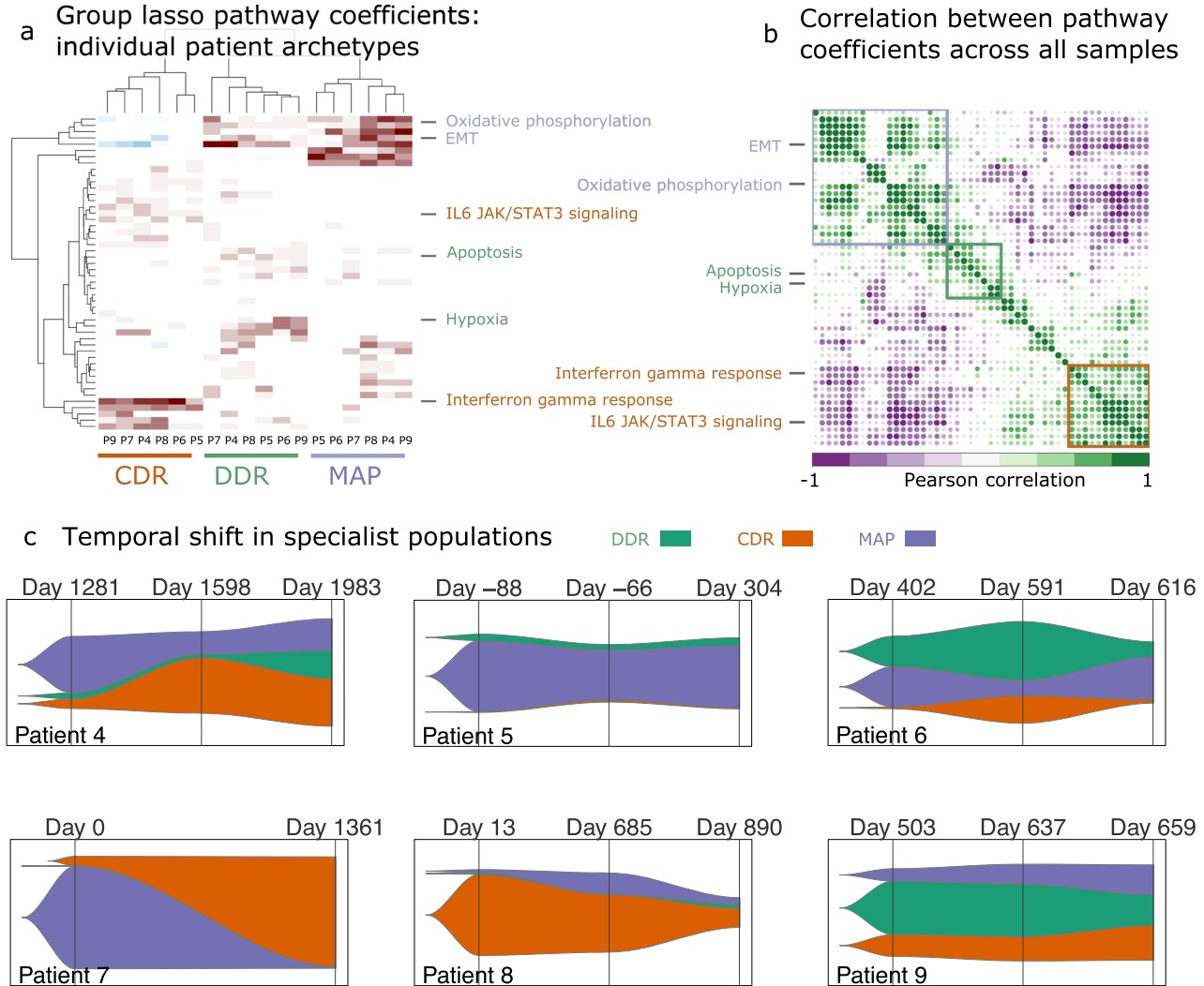

**Fig. 3 Archetype evolution in individual patients. a** Hierarchical clustering of group-lasso coefficients for archetypes determined for each longitudinal cohort patient across all time points. **b** Correlation plot of pathway scores across all samples with three major clusters indicated using colored boxes. Key pathways that defined the biological phenotypes associated with the archetypes in the integrated dataset are shown for both the hierarchical clusters and the correlation plots. **c** Fishplots displaying the temporal shift in the population of task specialists for each patient. The three major tasks are represented with their respected abbreviations (DDR DNA-damage repair, MAP metabolism and proliferation, and CDR cell defense response).

specialists and increase in MAP specialists in the post-treatment validation cohort suggested that multiple lines of chemotherapy may have also contributed to this shift.

To experimentally validate the observed shift towards the MAP archetype, we derived multiple primary cancer cells from patients 4 and 8 and tested the metabolic capacity changes over time. We created early passage primary patient cell lines using ascites samples from the two patients. In both cases, the late samples were obtained at an advanced stage after several lines of treatment. These serial cell lines displayed an increased basal ATP production capacity over time, with the majority of the energy production contributed by the glycolytic pathway in patient 4, and both oxidative phosphorylation and glycolysis in patient 8 (Fig. 4c). Lastly, to test the relative metabolic potentials in the cancer cells from our independent validation cohort, we also created cell lines from two pre-treatment patients (patients 16 and 17) and two post-treatment cohort patients (patients 21 and 23). We compared the ATP production rates for these four unmatched samples and found an overall increase in ATP production in the late treatment samples, contributed by both the glycolytic and oxidative phosphorylation pathways (Fig. 4d). The

total ATP production rates were elevated in the late time points in each study, while the relative contribution of oxidative phosphorylation and glycolysis also increased over time. We also evaluated the contribution of fatty acid oxidation by comparing the contribution to ATP production in the presence of a fatty acid oxidation inhibitor. We found a significant decrease in ATP production, but this reduction was consistent over time (Supplementary Figure 16).

**Temporal evolution of genomic variants in progressive HGSOC.** In order to test for association between genetic variants and archetype during tumor evolution, we next performed WGS analysis of germline and malignant samples from the longitudinal cohort patients. We observed an average of 12,000 SNVs and small indels in our samples along with an average tumor mutation burden of ~2.5/Mb, with six out of nine patients displaying an increase in mutation burden over time (Fig. 5a and Supplementary Data 7). An average of 800 structural variants (SVs) including indels (>25 bp) and breakpoints were observed in our samples (Fig. 5b and Supplementary Data 7). In addition, we also

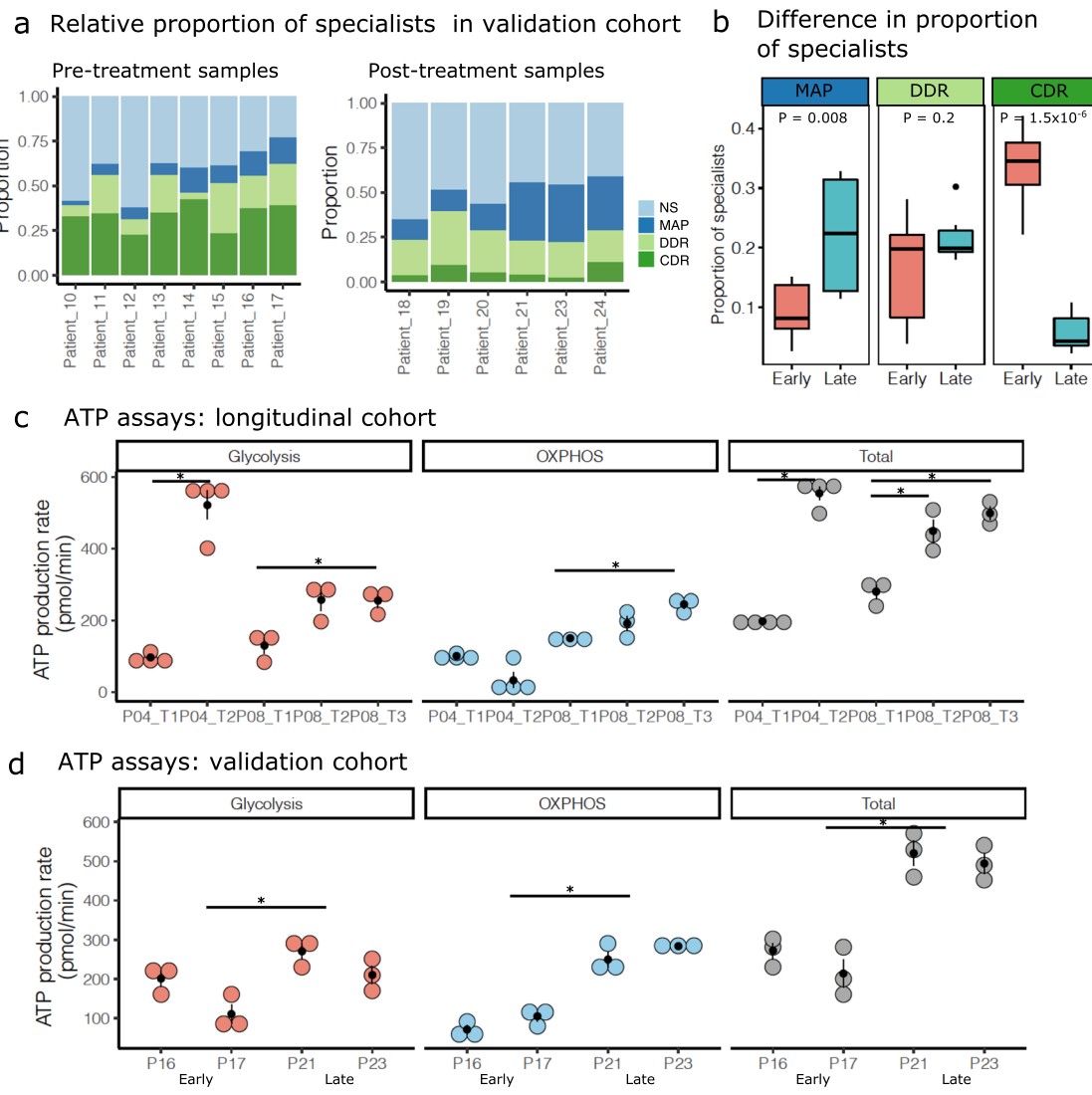

**Fig. 4 Archetype shifts and experimental validation of metabolic capacity. a** Stacked bar plots showing the proportion of specialists at biological tasks and non-specialists in the early/late patient cohorts. **b** Boxplots comparing the proportions of each task specialist between the early ($n = 8$) and late ($n = 6$) cohort samples. *P* value indicates significance in difference of means (two-tailed Student's *t* test). The lower and upper hinges in the boxplot indicate 25th and 75th percentiles and middle indicates 50th percentile (median), while the whiskers extended over 1.5× interquartile range. **c, d** Dot plots displaying glycolysis, oxidative phosphorylation (OXPHOS), and total ATP production in cell lines developed using the malignant cells isolated from patients 4 and 8 (**c**) and patients 16, 17, 21, and 23 (**d**). Vertical black bars indicate standard error of the mean (SEM) across three independent biological replicates. The horizontal bars with an asterisk indicate a significant difference (FDR < 0.05) in mean ATP production rates. The comparisons were performed for all time points in (**c**), and the joint means between early and late cohort patients in (**d**). Source data for ATP production assay are provided as Source Data file.

observed copy number gains or losses in one to eight cancer genes per sample (Fig. 5c). The non-synonymous SNVs, splice-site variants, indels, SVs (breakpoints), protein interaction variants, and copy number variants affecting cancer genes are shown in Fig. 5d.

To determine the potential pathogenicity of the non-synonymous SNVs and small indels, we searched for potential drivers by comparing the mutations in our samples with the IntoGen list of predicted and validated driver mutations. We also genotyped and predicted the impact of variants affecting HR genes in the germline samples. Pathogenic germline mutations in *BRIP1* in patient 1 and *BRCA1* in patient 8 were previously shown to contribute to the deficiency of the HR pathway[13,15]. Additional HR variants in germline samples were predicted to be either benign or common SNPs of unknown significance (Supplementary Data 8). Somatic missense and splice-site mutations in only one gene, *TP53*, were found to be potential

drivers. Truncal *EPHA3* mutations in patients 5 and 8 and *RHOA* in patient 9 were determined to be high impact based on SnpEff annotations; however, they were predicted as benign or passenger by IntoGen (Supplementary Data 9). We found frequent copy number gains of the *MYC* and *IGF2BP2* oncogenes, both associated with progressive ovarian cancers[32–34]. Similarly, copy number gains were observed in the *PIK3CA*[35], *ERBB2*[36], and *SOX2*[37] oncogenes, each reported to be potentially associated with chemotherapeutic resistance in ovarian cancers (Fig. 5d and Supplementary Data 10). In addition, we also observed a copy number loss of tumor suppressor genes *NF1* in patient 1 and *RB1* in patient 7 (Fig. 5d and Supplementary Data 10).

We next mapped the major acquired genomic events that tracked with the progression of the longitudinal cohort patients (Fig. 6). In patient 1, where the samples were profiled 1474 days apart, a copy number of loss of *NF1* along with a gain of *IGF2B2* were observed at time 2 (Fig. 6a). In the case of patient 2, we did

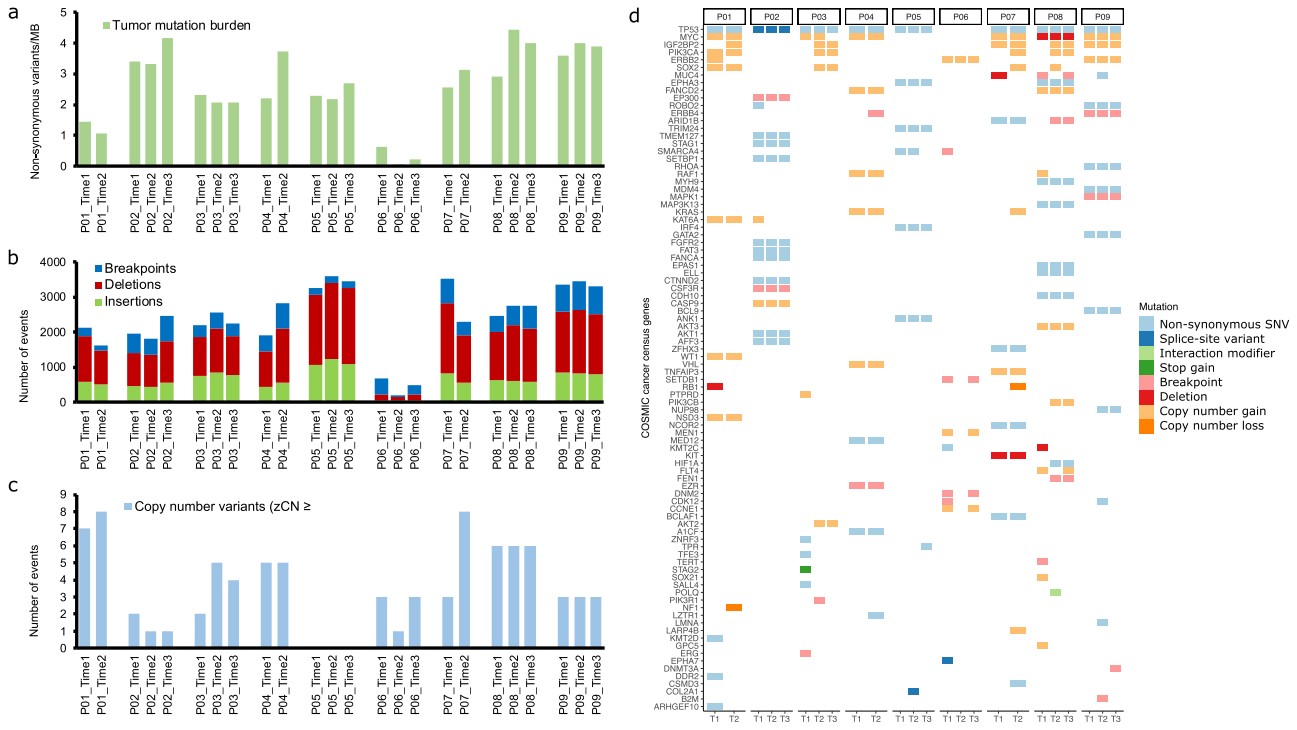

**Fig. 5 Overview of somatic WGS variants in the longitudinal cohort. a** Tumor mutation burden (non-synonymous SNVs and small indels/MB) in the longitudinal cohort samples. **b** Stacked bar plots showing the frequency of SVs, including indels (>25 bp) and breakpoints in each sample. **c** The number of genes with copy number alterations in each sample. The plot displays the number of genes that show a z-transformed copy number ≥2 (gain) or ≤−2 (loss). **d** Oncoprint displaying somatic variants affecting COSMIC cancer census genes. The plot displays SNVs and small indels with VAF > 0.05, large indels and breakpoints affecting exons, and CNVs with z-transformed copy number ≥2 (gain) or ≤−2 (loss).

not observe acquired copy number gains or losses in cancer genes. However, a breakpoint in the *ESR1* exon was acquired at the second time point (Fig. 6b). In the case of patient 3, the first sample was collected after the patient had already received first-line chemotherapy. We observed acquired copy number gains in *AKT2* oncogenes, along with a pathogenic *ABCB1-SLC25A40* fusion at the second time point (Fig. 6c) that had previously been reported.

Patient 4 did not show acquired CNVs at the second time point. Concurrent with an increase in the CDR archetype, we detected a passenger missense mutation in the *LTR1* exon and a breakpoint in the *ERBB4* exon of unknown significance (Fig. 6d). The CNV and SV profiles of patients 5, where the MAP was the critical archetype at all points, showed no variants that affected cancer genes. This patient acquired a splice-site *COL2A1* and missense *TPR1* passenger mutations at times 2 and 3, respectively (Fig. 6e). The SNV, CNV, and SVs in the samples from patient 6 profiled in a short period were truncal (Fig. 6f). Several key oncogenes were amplified in the second time point for patient 7, including *PIK3CA*, *KRAS*, and *SOX2*, along with a loss of *RB1* copy (Fig. 6g). These acquired driver mutations in patient 7 correspond to the emergence of CDR as the principal archetype at the second time point. *PIK3CA, PIK3CB*, and *IGF2BP2* were amplified at the second time point for patient 8, along with acquired breakpoints in the exons of *ARID1B* and *FEN1*. In contrast with patient 7, the second time point in the patient was associated with a decrease in CDR and an increase in the MAP archetypes. Overall, the relatively larger number of acquired events affecting similar pathways (PI3K/MAPK) in both patients 7 and 8 appeared to coincide with the long gap between the sample collection times but did not correspond to the evolution of the same archetypes (Supplementary Figure 17). Patient 9, where

the samples were profiled within a short time span and showed consistent enrichment of the CDR and MAP archetypes, did not acquire CNVs. However, a passenger missense variant in *NUP98* was identified at time 2 along with a breakpoint in an exon of *DMNT* at time 3.

To evaluate the association between key genomic variants and archetypes, we compared the proportion of specialists across samples grouped by the presence of a key mutation (Fig. 7). Samples grouped by pathogenic *TP53* mutations, present in 21 out of 24 samples, did not show any significant difference in the proportion of specialists for any archetype (Fig. 7a). Similarly, we did not observe a significant difference in the proportion of specialists in samples grouped by *MYC* gain in 14 samples (Fig. 7b), *IGF2BP2* gain in ten samples (Fig. 7c), *PIK3CA* gain in six samples (Fig. 7d), or *ERBB2* gain in seven samples (Fig. 7e). However, we found that *SOX2* gain detected in six samples was associated with a significantly higher proportion of CDR specialists (*P* = 0.02) (Fig. 7f). Other archetype specialists were not significantly different.

**Single-cell subclones are associated with emergent archetypes.** As shown above, driver genomic alterations that were acquired as a function of time or in response to chemotherapy could not completely explain the observed shift of all archetypes or development of therapeutic resistance over time. Therefore, we next investigated whether the subclonal architecture of the single cells might be associated with the emergence of the archetypes. We determined the subclonal structure of the longitudinal cohort scRNA-seq samples using the InferCNV method, assuming that the transcriptional heterogeneity at the single-cell level was driven by alterations that resulted in a change in expression levels of

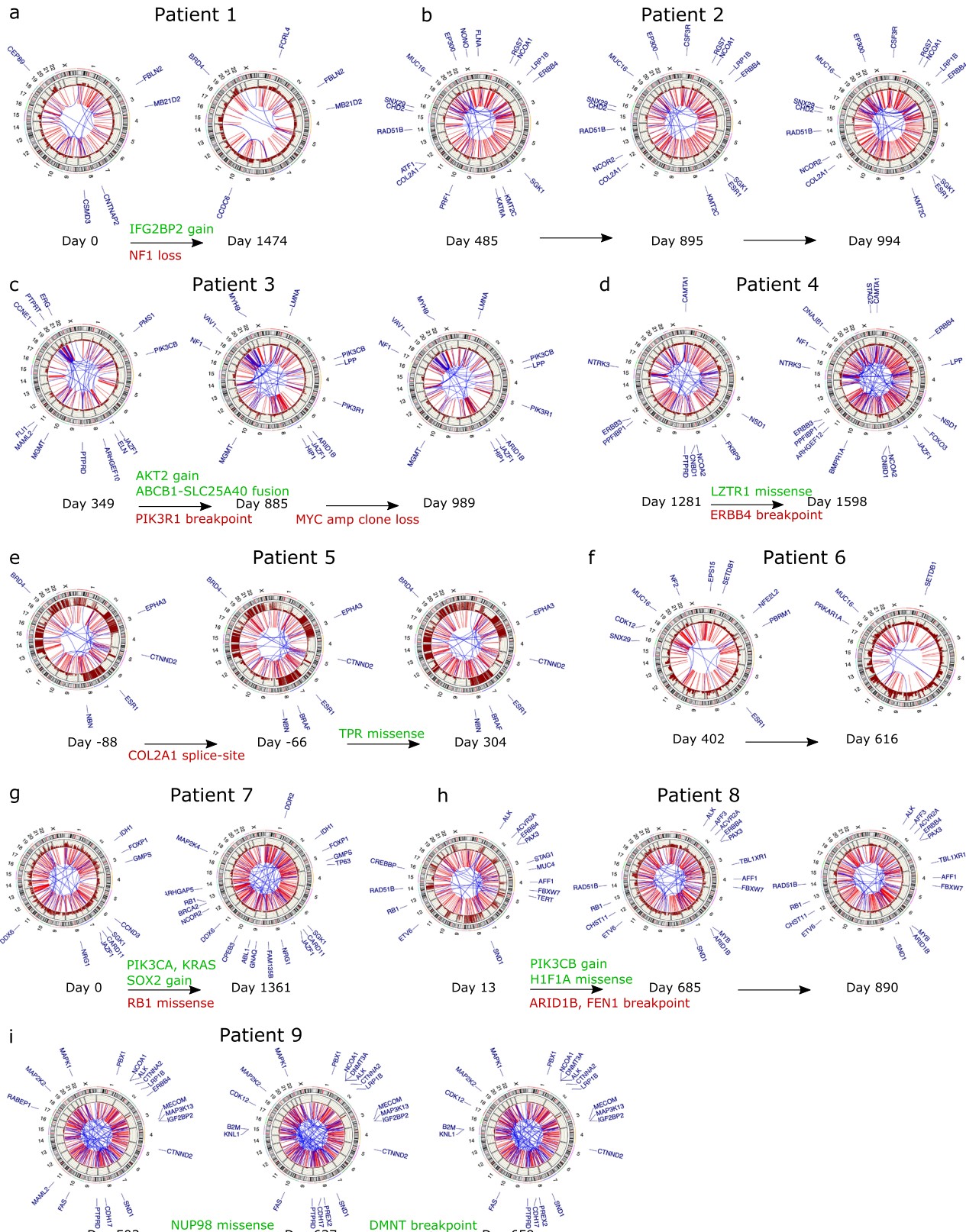

**Fig. 6 Temporal genomic characteristics of ovarian cancer. a–i** Circos plots displaying breakpoints and CNVs along chromosomes. The breakpoints are displayed in blue, if within chromosomes, or red, if between chromosomes. Copy number gains along the outer track are shown in dark red, with losses indicated in blue and neutral copies in dark gray. The annotations at the periphery of the plots for the first time point of each patient indicate CNVs and SVs (breakpoints) affecting the exons of COSMIC cancer census genes. The plots for the subsequent time points are annotated to show only acquired CNVs and SVs, with truncal breakpoints shown in light gray.

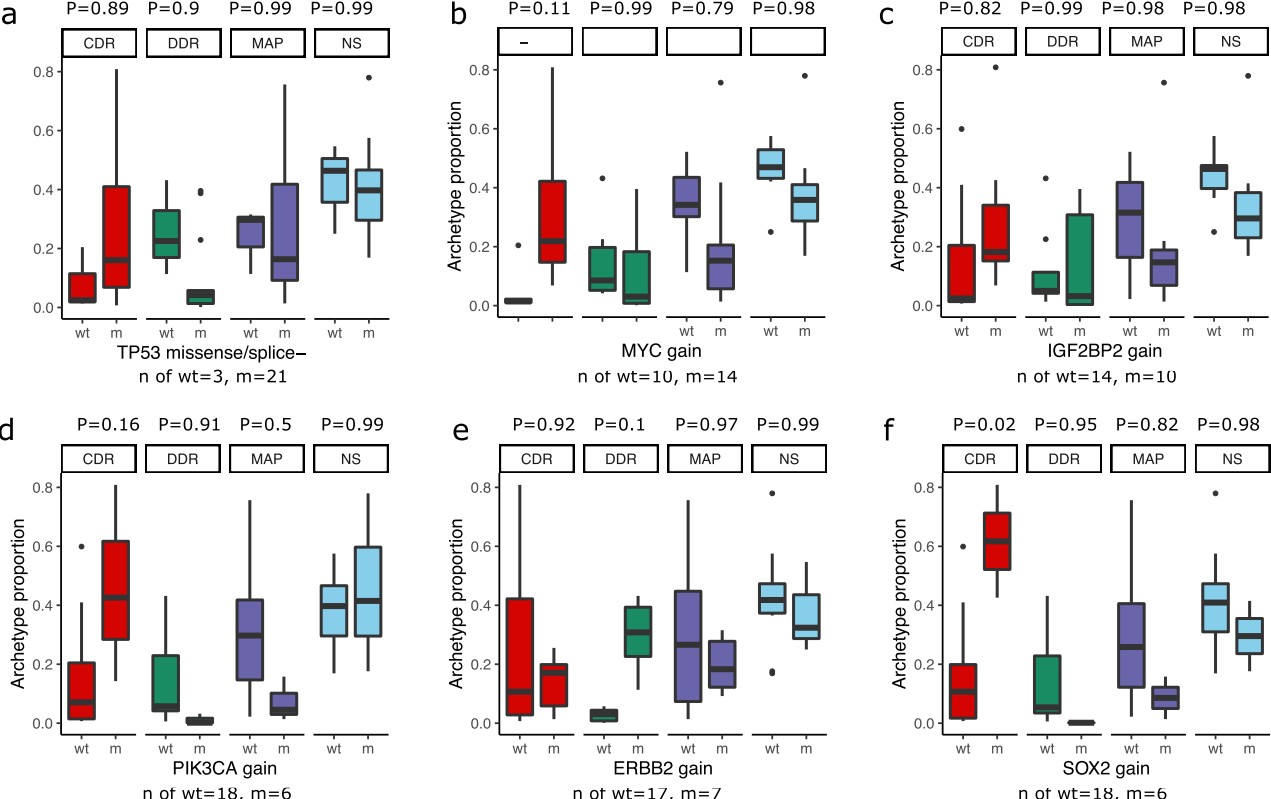

**Fig. 7 Archetype proportions in samples grouped by mutations in key genes. a–f** Boxplots showing the distribution of the proportion of each archetype specialist grouped by mutations that are considered pathogenic and appeared in at least 25% of the samples (6 out of 24). The boxes are grouped and colored by archetype, with red indicating CDR, green showing CDR, dark blue indicating MAP, and light blue showing non-specialists. The wt group label indicates samples with wild-type (or copy-neutral) gene, while m represents samples with a mutated (or amplified) gene with n of samples indicated at the bottom of each panel. The *P* value annotations above each archetype label indicate adjusted pairwise *P* values from Tukey's HSD (single-sided) post hoc analysis of the ANOVA model of the proportion of specialists with mutation status as an interaction term. The lower and upper hinges in the boxplot indicate 25th and 75th percentiles and middle indicates 50th percentile (median), while the whiskers extended over 1.5× interquartile range.

contiguous genes along the chromosomes[38] (Supplementary Figure 18). Overall, we found that the archetypes were significantly associated with specific inferred subclonal clusters in most patients (Tukey's honestly significant difference (HSD) *P* < 0.05) (Supplementary Figure 19).

The CDR archetype was associated with the sub-clone cluster 1 in patient 4, displaying an enrichment at the later time points compared to the initial time point (Fig. 8a), while the MAP archetype was associated with sub-clone cluster 2, and DDR with sub-clone cluster 3 (Supplementary Figure 19). In patient 5, the DDR archetype was associated with sub-clone cluster 1 and remained the core archetype throughout the study (Fig. 8b and Supplementary Figure 19). Patient 6 MAP archetypes enriched at later time points were associated with sub-clone cluster 3 (Fig. 8c and Supplementary Figure 13). The MAP archetype present in the first time point of patient 7 was enriched in the sub-clone cluster 3, while CDR that became the core archetype the later time point was linked to cluster 2 (Fig. 8d and Supplementary Figure 19). In patient 8, the MAP archetype was enriched at the later time points and tracked with sub-clone cluster 2 (Fig. 8e and Supplementary Figure 19). Patient 9 showed a shift towards the MAP archetype at the later time points and was associated with sub-clone cluster 3 (Fig. 8f). Thus, we observed that specific subclonal clusters were associated with the key MAP archetype in most patients. However, every cluster could not be mapped to an observed whole-genome amplification or deletion event. Therefore, further resolution of the genetic, epigenetic, and regulatory driver events behind the subclonal evolution of the HGSOC

tumors could help elucidate the mechanism of archetype shifts in these cells.

## Discussion

With the emergence of scRNA-seq technologies, it is now possible to study the patterns of transcriptional evolution at the cellular level. Understanding the patterns of transcriptional heterogeneity at the single cell level may help elucidate the mechanisms of chemoresistance in HGSOCs, especially in cases where driver single gene genomic alterations could not be detected. HGSOCs present a unique challenge, where the genetic heterogeneity is generally driven by SVs and CNVs, rather than single-gene driver mutations affecting cancer-related genes. This observation was confirmed in our WGS analysis of the temporal samples from the longitudinal cohort patients, where only truncal *TP53* driver mutations were detected across most patients.

Our study utilized the theory of multitask evolution to characterize transcriptional heterogeneity over time. This theory suggests that tumors perform various biological tasks; however, each cell within the tumor is optimized to perform only a specific task, which can evolve under selective pressure. By identifying the transcriptional task specialists in HGSOC cells and defining the core phenotypes that evolve during the course of treatment, we may be able to identify therapeutic targets against those critical phenotypes. A key finding of our study was the identification of three major archetypes or transcriptional specialists that could describe the gene expression of HGSOC cells. These included the MAP, DDR, and CDR archetypes (Fig. 2b, c). We found that the

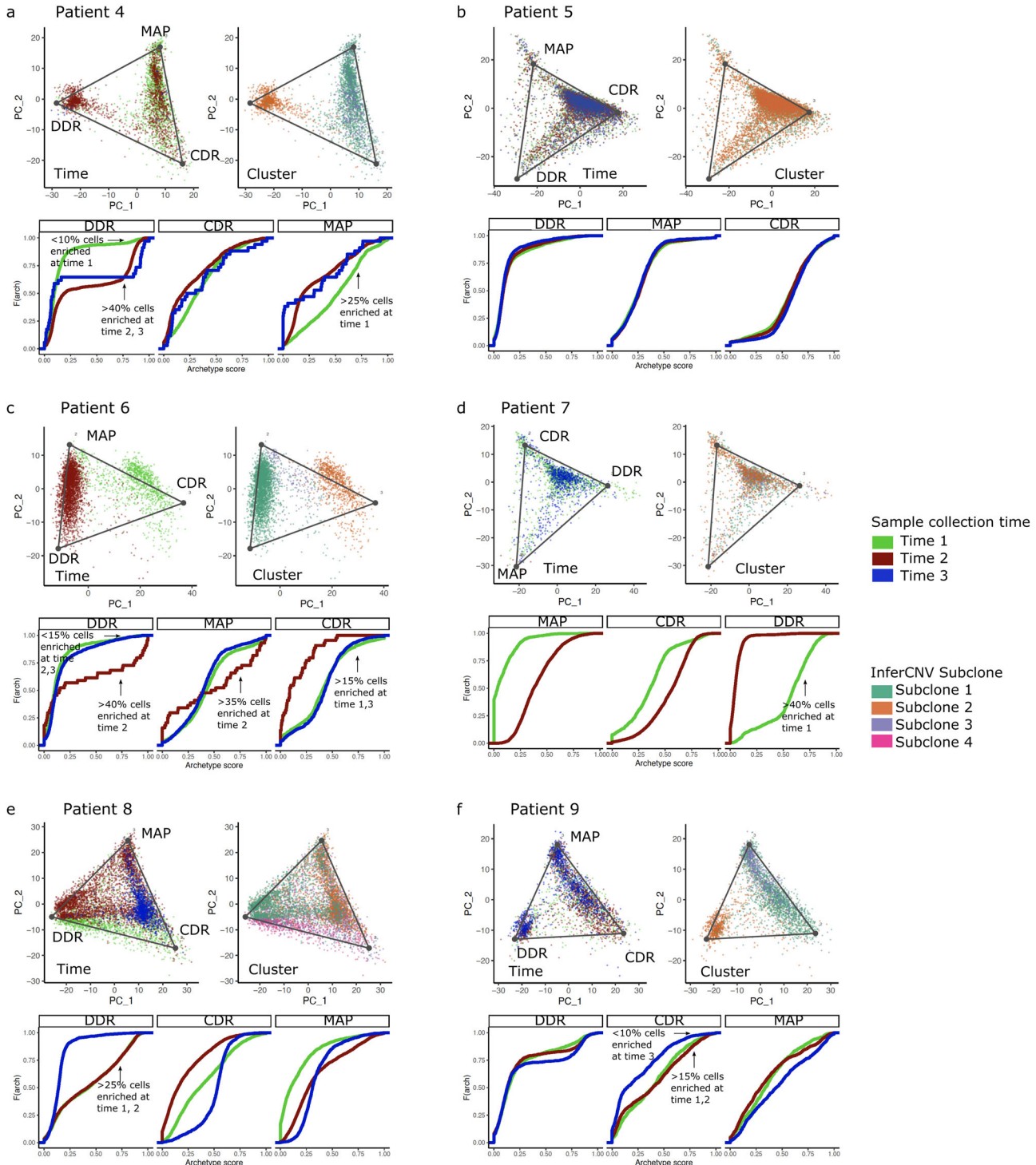

**Fig. 8 Mapping archetypes to time and inferred subclones. a–f** The top two panels display archetypes on the first two principal components of each longitudinal cohort patient. The panel on the left shows archetype projections on cells colored based on sample collection time, while the right panel shows cells colored based on clusters identified using the inferred CNV algorithm. The empirical cumulative distribution functions of the archetype scores at each time point. Shifts in enriched archetypes, defined by an archetype score >0.8, are annotated on the plots.

MAP archetype evolved later over the course of chemotherapy compared to early time points or treatment-naive samples and, at the last time point, this was concomitant with a decrease in the CDR archetype and correlated with poor overall survival of the patients (Supplementary Figure 15). Our results support the clinical observation that exceptional long-term HGSOC survivors are associated with enrichment of immune response signatures,

while short-term survivors tend to be associated with proliferation signatures[39].

Interestingly, pathways that are well known to contribute to cellular survival, metabolism, and proliferation from HGSOC bulk transcriptomes, such as MAPK[40] and WNT[41] signaling, were preferentially associated with nonproliferating cells in the CDR archetype, instead of the proliferating cells of the MAP archetype. The pathways enriched in the CDR archetype-like TLR

signaling and NK cell-mediated cytotoxicity suggest that these cells are responsive to immune cells in the microenvironment[42]. Thus, the prevalence of CDR cells could be indicative of active immune surveillance in the tumor, which has been linked to better prognosis and outcomes of ovarian cancers[43]. Key metabolic pathways (oxidative phosphorylation and glycolysis) were also associated with the CDR archetype (Supplementary Data 5). Based on this result, it is reasonable that a chemoresistant tumor would select metabolically active MAP cells that are actively proliferating, instead of metabolically active CDR cells that are subject to immune surveillance. This observation also supports the idea of multitask evolution, where the progressive tumors select for cells specializing in proliferation over immune response, assuming that both cell states have similar fitness costs as indicated by enrichment of metabolic pathways.

In previous reports, ovarian cancer cell lines have been characterized to show metabolic reprogramming of cancer cells that supported survival, promoted the development of chemoresistance, and disease progression[44,45]. Thus, our scRNA-seq models lend support to these in vitro observations by demonstrating a shift towards a high metabolism archetype in post-treatment patient-derived samples. Clinical interest to target this metabolic phenotype has garnered the attention of many investigators seeking to utilize combination therapies for more effective treatment options. In our WGS analyses, acquired SVs and CNVs affecting the *MYC* oncogene were detected in ~1/3 samples from the original nine patients of the longitudinal cohort. Increased *MYC* expression potentially contributes to the increased metabolic phenotype of ovarian cancers through increased glycolysis mediated by lactate dehydrogenase as well as glutamine addiction in MYC-driven cancers[46]. Thus, drugs such as BRD4 inhibitors that target the upstream pathways regulating *MYC* may be attractive candidates to control the metabolism and growth of chemoresistant cancer cells. Indeed, a small molecular BRD4 inhibitor, JQ1, has been shown to inhibit cell proliferation and induce apoptosis, as well as increase sensitivity to cisplatin in ovarian cancer cells[47,48].

In addition to the metabolic and proliferation pathways, we also observed a consistent emergence of the EMT pathway as one of the key hallmark predictors of the MAP archetype (Supplementary Figures 5, 6, 11, and 13). Previous studies have shown evidence linking EMT with both ovarian cancer progression and acquired chemotherapeutic resistance[49]. In addition, the activation of the EMT program is closely associated with increased plasticity, reprogramming of metabolism, and metastatic progression of cancer cells[50,51]. Recent studies show that the activation of the EMT program may be regulated via epigenetic mechanisms instead of somatic variants[52–54]. In addition, aberrant ovarian cancer cell metabolism was recently shown to be regulated by microRNAs using in vivo models[55]. Thus, the concurrent shift towards the MAP archetype and activation of the EMT pathway may be driven by concurrent epigenetic mechanisms beyond acquired driver mutations and serve as potential therapeutic targets[56].

A key outstanding question emerging from our study is the mechanisms that could explain the observed shift in archetypes over time. We found that driver somatic mutations were not associated with the emergence of archetypes across the patients. We have evaluated the association between the presence of key somatic mutations in tumor samples across all time points and the likelihood that a particular archetype might be enriched in a tumor carrying the somatic mutation (Fig. 7). While most of the common mutations are not associated with a particular archetype, it is possible that the emergence of the archetypes can be explained by other somatic mutation changes that are not yet characterized to be associated with HGSOC progression. As our study is underpowered to discover new somatic variants, our results do not completely rule out the potential role of genetic mechanisms in archetypal evolution, as evidenced by the close association of archetype shifts with specific subclones. However, resolving the exact subclonal structure and determining specific somatic mutations in single-cell subclonal populations is quite challenging due to the low depth of coverage and sparsity of the scRNA-seq data. Improvements in scDNA-seq technologies and the development of analytical methods to resolve somatic mutations in single cells may help bridge this gap in knowledge[57]. On the other hand, transcriptional evolution could also be driven by nongenetic mechanisms, including epigenetic alterations and acquired changes in the noncoding transcriptome of single cells[58,59]. Complete characterization of such mechanisms would require new technologies to simultaneously profile and study such changes.

Overall, our study shows the existence of three core archetypes in HGSOCs. We found that the existence of the three archetypes and the biological phenotypes associated with each archetype is remarkably consistent across patients. These results suggest a common underlying biological theme exists across HGSOCs, despite the variations in the baseline tumor genomes arising from the inherent chromosomal instability related to loss of DNA-damage repair mechanisms[60]. This inherent commonality or phenotypic stability of the HGSOCs could perhaps suggest genetic canalization, which also explains the exceptional robustness of the disease against treatment[61]. Thus, with the apparent ability of tumors to evolve via nongenetic mechanisms[58,62], it may ultimately require the development of novel treatment strategies that simultaneously target the genomic and phenotypic vulnerabilities that evolve in the HGSOC tumors. Our study provides compelling evidence of HGSOC evolution to be associated with a shift towards a high-MAP state, with a concomitant decrease in immune response state, after receiving multiple lines of chemotherapy in two separate patient cohorts. In vitro assays confirm the shift towards a high-metabolism state in the post-treatment samples. Taken together, these results suggest the potential of new avenues of therapy that specifically target the metabolic pathways to overcome chemoresistant HGSOCs.

## Methods

**Sample collection and processing.** Ascites or pleural effusions were drained and collected from nine ovarian cancer patients longitudinally over the course of patient treatment. Samples were collected with proper written informed consent and ethical compliance under IRB # 07047 an 17334 (City of Hope), 41030 and 89989 (University of Utah), or HREC # 01/60 and 16/161 by the Australian Ovarian Cancer Study (AOCS), which were analyzed under HREC # 15/84 (Peter MacCallum Cancer Centre). Malignant fluids were centrifuged at $500 \times g$ for 5 min to pellet cells. Red blood cells were removed by lysis in Tris-ammonium chloride buffer (17 mM Tris, pH 7.4, 135 mM ammonium chloride) and incubated for 5 min in a 37 °C water bath. Cells were then centrifuged at $500 \times g$ for 5 min at room temperature and repeated until red blood cells were removed. Cells were washed in 1× phosphate-buffered saline (PBS) (Gibco, Cat. # 10010) before frozen viably in 50% RPMI-1640 (Gibco, Cat. # 11875) + 40% fetal bovine serum (FBS, Sigma, Cat. # 12306C) + 10% dimethyl sulfoxide (DMSO) (Fisher Scientific, Cat. # D2650). Ascites fluid collected by the AOCS was centrifuged at $450 \times g$ for 10 min at 4 °C. Red blood cells were removed by incubation in ice-cold lysis buffer (14.4 μM NH₄Cl, 1 μM NH₄HCO₃) at room temperature for 10 min. Cells were centrifuged at $450 \times g$ for 10 min at 4 °C, washed in 10% FBS in 1× PBS, and centrifuged again. Cells were frozen viably in FBS + 10% DMSO. One sample (Patient 22) was dropped from further analysis upon receiving an updated classification of this sample as a high-grade solid endometrial tumor.

**Cancer cell isolation.** Frozen viable ascites or pleural effusion cells were thawed, centrifuged at $300 \times g$, and resuspended in 1× PBS to determine the concentration, viability, and cancer cell purity by trypan blue staining. In some cases, cancer cells were purified by Miltenyi Biotec QuadroMACS by negative selection of CD45⁺ (CD45 MicroBeads, Miltenyi Biotec, Cat. # 130-045-801), CD90⁺ (CD90 MicroBeads, Miltenyi Biotec, Cat. # 130-096-253), and podoplanin-expressing cells (biotinylated anti-podoplanin antibody, BioLegend, Cat. # 337015 and Miltenyi Biotec Anti-Biotin MicroBeads Cat. # 130-105-637). Cells were first labeled using

20 μL of anti-podoplanin antibody in 500 μL 1× PBS + 0.5% BSA (bovine serum albumin, EMD Millipore, Cat. # 12661525) per $10^7$ cells, incubated 10 min at 4 °C, washed with 2 mL 1× PBS + 0.5% BSA, centrifuged at 300 × g for 5 min, then resuspended in 60 μL of 1× PBS + 0.5% BSA per $10^7$ cells. Then, 26 μL of each CD45, CD90, and Anti-Biotin Microbeads were added to cell suspension (per $10^7$ cells) and CD45+-, CD90+-, and anti-podoplanin biotin-labeled cells were depleted using LD columns (Miltenyi Biotec, Cat. # 130-042-901) according to the manufacturer's instructions. The samples were processed using the StemCell EasyEights EasySep column-free magnet to remove CD45+ (EasySep CD45 Depletion Kit II, Cat. # 17898), and/or dead cells (EasySep Dead Cell Removal (Annexin V) Kit, Cat. # 17899) as appropriate. To isolate cancer cells using StemCell EasySep Antibody Kits, cells were centrifuged and resuspended in 1× PBS + 2% FBS + 1 mM calcium chloride (G-Biosciences, Cat. # R040) to a concentration of <$10^8$ cells per 2 mL total volume and transferred to a round bottom 5 mL FACS tube. Sequentially Dead Cell Removal Cocktail (50 μL/mL sample) and Biotin Selection Cocktail (50 μL/mL sample) were added and incubated at room temperature for 3 min, followed by CD45 Depletion Cocktail (50 μL/mL sample) and incubated at room temperature for 5 min. StemCell RapidSphere magnet beads were added (75 μL/mL for CD45 RapidSpheres and 100 μL/mL for Dead Cell RapidSpheres) and incubated at room temperature for 3 min off the magnet. Cell samples were then incubated on EasyEight magnet for 5 min, collected supernatant, and repeated additional Easy-Eight magnet column cleanup. Collected cells were then centrifuged and resuspended in 1× PBS and maintained at 4 °C.

**Nuclei isolation**. After cancer cell isolation, patient samples that did not dissociate into single-cell suspensions or had a high proportion of cancer cell clusters were then processed to isolate single-nuclei suspensions. To isolate nuclei, cells were resuspended in (4:1) Lysis Buffer (10 mM Tris-HCl, pH 7.8 (Teknova, Cat. # T1078), 146 mM NaCl (Alfa Aesar, Cat. # J60434AK), 1 mM CaCl₂ (G-Biosciences, Cat. # R040), 21 mM MgCl₂ (G-Biosciences, Cat. # R004), 0.05% BSA (EMD Millipore, Cat. # 12661525), 0.2% Igepal CA-630 (MP Biomedicals, Cat. # 198596), DNase/RNase-free water (Gibco, Cat. # 10977)):DAPI buffer (106 mM MgCl₂, 50 μg/mL 4′, 6-diamidino-2-phenylindole (DAPI, Invitrogen, Cat. # D1306), 5 mM ethylenediaminetetraacetic acid (EDTA, Quality Biological Inc., Cat. # E522100ML), DNase/RNase-free water)) supplemented with fresh 0.2 U/μL SUPERase·In RNase Inhibitor (Invitrogen, Cat. # AM2694). Cells were incubated for 15 min at 4 °C to release nuclei. The lysate was then filtered through a 40 μm mesh filter (Falcon, Cat. # 352340) collecting nuclei in flow through. All downstream nuclei processing utilized Eppendorf LoBind DNA tubes to prevent nuclei loss. Nuclei were centrifuged 500 × g, at 4 °C, for 5 min and washed two times with 500 μL of 1× PBS + 1% BSA + 0.2 U/μL SUPERase·In RNase Inhibitor. Nuclei were resuspended in 1× PBS + 1% BSA + 0.2 U/μL SUPERase·In RNase Inhibitor at a target of 1000 cell/μL, re-filtered using a 40 μm mesh filter, and counted on a hemocytometer by DAPI fluorescence using an Invitrogen Countess equipped with DAPI filter cube and maintained at 4 °C.

**Single-cell RNA-sequencing**. scRNA-seq was performed on single-cell or single-nuclei suspensions using either the Takara Bio ICELL8 Single-Cell System or the 10X Genomics Chromium to prepare cDNA sequencing libraries. Samples processed on the ICELL8 Single-Cell System (Takara Bio) were prepared using the SMARTer ICELL8 3′ DE Reagent Kit V2 (Takara Bio, Cat. # 640167) from isolated nuclei. DAPI-stained nuclei were diluted to a concentration of 60,000 cell/mL in 1× PBS + 1% BSA + 1× Second Diluent + 0.2 U SUPERase·In RNase Inhibitor and dispensed onto the ICELL8 3′ DE Chip (Takara Bio, Cat. # 640143) using the ICELL8 MultiSample NanoDispenser. Single-nuclei candidates were selected using the ICELL8 Imaging System with ICELL8 CellSelect Software (Takara Bio, V1.1.10.0) selecting for DAPI-positive nuclei and reverse transcription, and sequencing library preparation was performed according to the manufacturer's instructions. ICELL8 cDNA sequencing libraries were sequenced at a depth of 200K reads per cell on Illumina HiSeq 2500, read #1 = 26 nt and read #2 = 100 nt.

Samples processed on the 10X Genomics Chromium were processed using the Chromium Single Cell 3′ V3 Kit (10X Genomics, Cat. # 1000075) using whole cells or isolated nuclei. Single cells or nuclei were diluted to a target of 1000 cell/μL in 1× PBS (whole cells) or 1× PBS + 1.0% BSA + 0.2 U/μL SUPERase·In RNase Inhibitor to generate GEM's prepared at a target of 5000 cells per sample. Barcoding, reverse transcription, and library preparation were performed according to manufacturer instructions. 10X Genomics generated cDNA libraries were sequenced on Illumina HiSeq 2500 or NovaSeq 6000 instruments using 150 cycle paired-end sequencing at a depth of 10K reads per cell. scRNA-seq was performed at the Integrative Genomics Core at City of Hope, Fulgent Genetics, and the High Throughput Genomics Core at Huntsman Cancer Institute (HCI) of the University of Utah.

**Genomic DNA isolation and WGS**. Genomic DNA was isolated using the QIAamp DNA Micro Kit (Qiagen, Cat. # 56304) according to the manufacturer's instructions for isolated cancer cells and nuclei suspensions from scRNA-seq, as well as patient-matched buffy coat for germline DNA. Germline DNA was also isolated from patient-matched isolated peripheral lymphocytes using the salting-out method. Briefly, lymphocytes were resuspended in nuclei lysis buffer (0.1 M Tris pH 8, 2 mM EDTA pH 8, 0.1 M NaCl, proteinase K, and sodium dodecyl

sulfate), and incubated at 56 °C for 1 h followed by 37 °C for 3 h. Saturated salt solution (~6 M NaCl) was added to lysed cells, which were centrifuged at 18,400 × g for 15 min at 4 °C after vigorous mixing. The supernatant was transferred to ice-cold 100% ethanol and the tubes were rocked gently until the DNA precipitated. After overnight incubation in ethanol at −20 °C, DNA was rinsed twice by placing in 70% ethanol, centrifugation, and removing the ethanol. DNA was air-dried and resuspended in sterile water. WGS DNA libraries were prepared using either NEBNext Ultra II DNA Library Prep Kit (New England Biolabs), KAPA Hyper Prep PCR Free Library Prep Kit (Roche), or Nextera DNA Flex Library Prep Kit (Illumina), and sequencing performed on Illumina NovaSeq 6000 instruments at 150 cycles and paired-end sequencing to read depth of 40–60× coverage. Sequencing was performed at Admera Health, Fulgent Genetics, and the High Throughput Genomics Core at HCI of University of Utah.

**Cell culture**. To create stable patient-derived primary cell lines, frozen patient ascites were processed and then immediately placed in media as specified below. All cells were maintained in RPMI-1640 (Gibco; Cat. # 11875085) supplemented with 10% heat-inactivated FBS (Sigma, Cat. # 12306C) and 1% antibiotic/anti-mycotic solution (Gibco; Cat. # 15240062) in uncoated filter top polystyrene flasks and maintained at 37 °C in 5% CO₂, patient cells were additionally kept in 5% O₂ hypoxic simulated humidified air.

**Metabolic assays**. ATP production rates were assayed with the XF Real-Time ATP Rate Assay Kit (Agilent, Cat. # 103592-100) as per the manufacturer's instructions. Briefly, cells were plated down in the Seahorse XF96 cell culture microplates at 10,000 cells/well/80 μL and placed back in 37 °C, 5% CO₂ incubator. After 24 h, cells were washed in assay media made up from Seahorse XF RPMI Media, pH 7.4 (Agilent, Cat. # 103576-100) containing 10 mM glucose (Agilent, Cat. # 103577-100), 1 mM pyruvate (Agilent, Cat. # 103578-100), and 2 mM L-glutamine (Agilent, Cat. # 103579-100) and incubated for 1 h in a non-CO₂ incubator at 37 °C before a final wash in the assay media. The Seahorse XFe96 analyzer was calibrated and the assay was run using a standard XF Real-Time ATP Rate template created using the WAVE Software (V2.6.1) and assay standard drug injections were used of 1.5 μM oligomycin in port A and 0.5 μM rotenone/anti-mycin A in port B.

Results for each well were normalized by cell count using 1 μg/mL Hoechst that was added to port B with the rotenone/antimycin A cocktail and injected automatically, then visualized by imaging the wells at ×4 on the Cytation5 Multimode Cell Imager (BioTek) and analyzed with the GEN5 Software (BioTek; V3.0.5) for cell count. If multiple plates were needed for comparison, OAW42 cells were plated down at 5000 cells per well in triplicate 24 h before the assay for environmental variable normalization between plates. Analysis for the ATP rate assay was performed using the Agilent ATP report generator as per the manufacturer's recommendations.

Basal oxygen consumption rates were determined with the Seahorse XF substrate oxidation assay, by using the XF Long-Chain Fatty Acid Oxidation Stress Test Kit (Agilent, Cat. # 103672-100) as per the manufacturer's instructions. Briefly, 3000 cells were plated down in RPMI-1640 + 10% heat-inactivated FBS and 1% antibiotic/antimycotic solution cell culture media, and on the day of the assay, the cells were washed with defined Seahorse XF RPMI Assay Media, pH 7.4 (Agilent, Cat. # 103576-100) containing 10 mM glucose (Agilent, Cat. # 103577-100), 1 mM pyruvate (Agilent, Cat. # 103578-100), and 2 mM L-glutamine (Agilent, Cat. # 103579-100). The fatty acid oxidation inhibitor, Etomoxir, was resuspended to a final assay concentration of 4 μM, and either Etomoxir or defined media were injected automatically through port A during the Seahorse substrate oxidation assay run on the XFe96 analyzer and controlled by the WAVE Software (V2.6.1). Metabolic rates for each well were normalized to the number of cells in each well using Hoechst (1 μg/mL). Analysis was performed using the Agilent analytic's cloud-based analysis tool (https://seahorseanalytics.agilent.com/) as per the manufacturer's recommendations.

**Cell growth and viability assays**. Cell viability of the matched samples from patient 4 (two samples) and patient 8 (three samples) was assessed by the CellTiter-Glo 2.0 cell Viability Assay (Promega; Cat. # G9241) as per the manufacturer's instructions. Briefly, 1000 cells per well were plated in triplicate, clear-bottom 96-well plate in RPMI-1640 + 10% heat-inactivated FBS and 1% antibiotic/anti-mycotic solution cell culture media. After 12 days, the cells were equilibrated to room temperature for 30 min and then equal volumes of the CellTiter-Glo reagent was added to each well and placed on an orbital shaker for 2 min, then allowed to incubate for a further 10 min at room temperature, and luminescence was read on a plate reader (Tecan Infinite M1000). The growth of the cells in the 96-well plates was also assessed by imaging each well every 24 h in a Cytation5 Multimode Cell Imaging System (BioTek). Specifically, a phase-contrast image was taken with a 4 × 4 montage, and then the GEN5 Software (BioTek, V3.0.5) was used to stitch the image together and cell analysis calculated the cell count of each well. Both cell growth and viability were plotted with GraphPad (Prism V8.4.3).

**scRNA-seq analysis**. Raw scRNA-seq data were preprocessed in the Bioinformatics ExperT SYstem (BETSY)[63] using the Cell Ranger v2.1.1 pipeline for 10×

data, aligned to the hg19 reference genome using the STAR aligner[64], followed by extraction of read counts using featureCounts[65]. The resulting count matrix of cells was used for downstream analysis using the R package Seurat v3[66]. High-quality cells were identified based on the following criteria: a minimum of 1000 total number of expressed genes per cell, a minimum of 2000 UMIs per cell, and a percentage of mitochondrial genes <25%. Count matrices from individual patient samples were normalized and integrated using the CCA algorithm for batch correction[66]. This was followed by principal component analysis of the variable genes in the integrated dataset, clustering using unsupervised graph-based clustering, and dimensionality reduction using uniform manifold approximation (UMAP) or $t$-distributed stochastic neighbor embedding.

The cell-type identities of the clusters were determined using a two-step approach. A first pass prediction was performed using the SingleR reference-based classification approach[24] using references based on the ENCODE[67] and HPMC[68] datasets. Next, the individual markers corresponding to predicted cell types were mapped on to the clusters to confirm their classification. In addition, we classified malignant epithelial cells and normal cells by inferring chromosomal copy number aberrations from the scRNA-seq data using the method by Patel et al.[17] (see Supplementary Methods for details). The copy numbers were inferred using the R package InferCNV, using predicted fibroblasts as reference. For pathway enrichment, raw counts were first normalized using the method proposed by Rizzo et al.[69]. Then, a single sample gene set enrichment scores were calculated for hallmark[70] and curated molecular signature[71] gene sets using the GSVA package for R[72].

**WGS analysis**. Germline and tumor WGS sequencing raw reads were pre-processed using the BETSY to add read-groups, mark duplicates, perform indel realignment, base quality recalibration, sorting and indexing, and alignment to the hg19 reference genome using BWA MEM to generate BAM files. Allele-specific CNVs calls, along with ploidy and cellularity estimates, were called from the BAM files using Sequenza[73] or Facets[74] CNV callers using the corresponding germline BAM files of that patient as reference. For each sample, the CNV calls were z-transformed (allele-specific copy number − mean sample copy number/standard deviation of sample copy number) and rounded to the nearest integer for comparison. Copy number alterations were defined as z-transformed copy numbers of ≥2 for gains and ≤−2 for losses.

Germline variants in homologous repair genes (*ATM, ATR, CHEK1, CHEK2, BRCA1, BRCA2, BARD1, BRIP1, FAM175A, MRE11A, NBN, PALB2, RAD51C, RAD51D*) along with *TP53 and RB1* were determined by genotyping the germline BAM files using GATK, platypus, varscan, and freebayes. Variants detected by at least two callers and with a variant allelic frequency ≥ 0.05 were retained and annotated using SnpEff[75] to determine non-synonymous variants. Somatic SNVs and small insertions or deletions were determined from the BAM files using strelka[76], mutect2[77], and muse[78] variant callers. Genes with a variant allele frequency ≥ 0.05 determined by at least two callers were retained for further analyses after adjusting for cellularity as determined from the CNV callers. Non-synonymous variants were first determined using SnpEff. Cancer genes were defined based on the list of cancer census genes from COSMIC[79]. Potential driver mutations were defined based on the list of known or predicted drivers in the IntoGen database[80]. SVs, including insertions, deletions, and breakpoints, were called and annotated using SvABA[81]. CNV and SVs were visualized as circos plots using the R package RCircos[82].

**Archetype analysis and biological task classification**. We analyzed the HGSOC scRNA-seq transcriptomes intending to identify distinct biological tasks that each of the cells needs to perform and face evolutionary trade-offs[25]. Based on the theory proposed by Shoval et al.[22], we seek to represent the transcriptome datasets as a Pareto-optimal situation by identifying that encloses the data with the vertex of the polytope representing task-specific archetypes. For this analysis, we used the first five principal components of the CCA-normalized scRNA-seq data from the longitudinal cohort, individual patient samples from the longitudinal and the early (treatment-naive), or late (multiline treatment) cohorts. We used the ParetoTI package for R (https://github.com/vitkl/ParetoTI) to determine the minimum number of vertices required to enclose the transcriptome data based on the principal convex hull algorithm[83]. Fitting polytopes with an increasing number of vertices ranging from 3 to 8 revealed three vertices (triangle) were sufficient to enclose the data in each case, with additional components resulting in no gain in the proportion of variance explained by the resulting polytope. In addition, we analyzed the CCA-normalized count data to verify the number of archetypes using the ParTI package for MATLAB[25]. We calculated the variance explained by increasing the number of archetypes and confirmed that the three archetypes were ideal using the elbow method. A $t$-ratio test ($P < 0.01$) confirmed that the polytope was a statistically significant fit for the data. Subsequently, the polytope fit and archetype scores, or standardized Euclidean distance of each cell to the nearest vertex, were determined using the ParetoTI package. For each archetype, specialist cells were defined as cells above the 80th percentile of archetype scores, while cells that did not meet this criterion for any archetype were classified as non-specialists. The evolution of the archetypes was represented as the percentage of specialists at each time point using the R package fishplot[84].

To determine the biological tasks that described each archetype, we used a gaussian multitask or multinomial model with the set of archetype scores as the outcome variable and the hallmark gene set enrichment scores or CCA-normalized gene expression of each cell as the set of predictors. The multitask model was fit using a group-lasso penalty using the R package glmnet[85]. Briefly, 10-fold internal cross-validation was performed with a lasso penalty ($\alpha = 1$) to determine the multitask model error over varying penalty parameter ($\lambda$) values. The contribution (coefficients) of each pathway to the model based on the fraction of deviance explained was also assessed over varying levels of degrees of freedom. Top pathway phenotypes contributing to the model were used to define the phenotypes associated with each archetype. Subsequently, the model was fit using a $\lambda$ value within one standard error of the minimum. The group-lasso coefficients of each hallmark pathway were then analyzed using hierarchical clustering and correlation analyses to determine clusters of related pathways that were associated with each archetype. Further, the identities of the archetypes were validated based on repeated clustering patterns of the pathway coefficients determined using multitask learning analysis of the individual patient archetypes, co-clustering of coefficients from related pathways, and expression levels of key genes that were available in the normalized scRNA-seq dataset.

**Reporting summary**. Further information on research design is available in the Nature Research Reporting Summary linked to this article.

## Data availability

Single-cell RNA-seq data generated and analyzed during this study are available from the GEO database under accession GSE158722.

Whole-genome sequencing and raw scRNA-seq data are available under controlled access from dbGaP under the accession ID phs002294. Catalog of driver genes from IntoGen is publicly available (https://www.intogen.org/download?file=IntOGen-Drivers-20200201.zip). Data associated with the ATP assays and *P* values from multiple comparisons analyzed using Tukey's HSD test are provided with this paper. The remaining data are available within the Article, Supplementary information, or from the authors upon request. Source data are provided with this paper.

## Code availability

The BETSY software environment[63] used for the bioinformatic analyses is available at https://github.com/jefftc/changlab. Custom pipelines for the preprocessing of scRNA-seq, WGS, and gene set enrichment analyses with BETSY, and Seurat, archetype, and multitask learning analyses with R are available at https://github.com/U54Bioinformatics. Analyses with R-packages were performed in R-Studio (version 1.2.5033; R version 3.6.3.).

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

## Acknowledgements

This study was supported by a US National Cancer Institute U54 grant (U54CA209978) awarded to A.H.B., J.T.C., and D.D.L.B. D.D.L.B. is supported by National Health and Medical Research Council of Australia (NHMRC) grants APP1092856 and APP1117044. E.L.C. is supported by NHMRC grants APP1124309 and APP1161198. The Australian Ovarian Cancer Study (AOCS) was supported by the US Army Medical Research and Materiel Command under DAMD17-01-1-0729. We wish to thank Fred Adler, Mark Smithson, and members of the City of Hope U54 community for their invaluable feedback and support.

## Author contributions

Conceptualization: A.N., A.H.B., and D.D.L.B. Data curation: P.A.C., E.L.C., S.M., J.T.C., and L.P. Formal analysis: A.N., P.A.C., H.M., B.C., L.P., and A.H.B. Funding acquisition: A.H.B., D.D.L.B., and J.T.C. Investigation: A.N., H.M., B.C., and P.A.C. Methodology: A.N., A.H.B., J.T.C., and L.P. Project administration: A.H.B., P.A.C., and D.D.L.B. Resources: M.C.C., E.S.H., S.J.L., E.W.W., S.F., N.T., R.S., T.W., A.L.C., P.M., A.H.B., and D.D.L.B. Software: J.T.C., A.N., H.M., and L.P. Supervision: A.H.B. and D.D.L.B. Visualization: A.N. and H.M. Writing—original draft: A.N., P.A.C., and B.C. Writing—review and editing: A.N., A.H.B., D.D.L.B., E.L.C., and J.T.C.

## Competing interests

The authors declare no competing interests.
