## [Peer Review File · Nature Communications]

REVIEWER COMMENTS

Reviewer #1, expert in high-grade serous ovarian cancer, metabolism and drug resistance (Remarks to the Author):

The authors sought to determine the key phenotypic transcriptomic changes that evolve during chemotherapy due to selective pressure. They employed the Pareto optimization concept, to determine the tasks that dominate an organism's fitness and identified a number and features of driver phenotypes "archetypical phenotypes" in cancer cells that evolve during tumor progression and therapy. The authors also evaluated the archetypes with genetic alterations to identify the potential link between somatic alterations and phenotypic state. The authors identifies 3 archetypes that characterize cancer cell phenotypes during cancer progression and chemotherapy. These included metabolism and proliferation (MAP), cellular defense response (CDR), and DNA damage repair (DDR). The manuscript is well written with comprehensive description of the methodology and analysis. However, I have the following concerns

1. The sample size, though longitudinal over years is small and some patient variabilities cannot be generalized or linked to as specific somatic or germ-line mutation. In addition, the prognostic power of the findings to predict disease outcome (responders or resistance) to a given treatment, or development or recurrence is not addressed.
2. The authors stated that the genes included in the second archetype cell defense response (CDR) include PI3K/AKT and MTORC1. These signatures are better included with Metabolism and proliferation (MAP) archetype as they represent the main pathways linking cell proliferation to metabolic programming. The authors should revisit their statements and re-assign these pathways.
3. In lines 196-198, there is discrepancy of patients/ samples numbers of the validation cohort.
4. While the data from the scRNA of longitudinal samples in the pilot and validation cohorts identified malignant cell tendency to cluster into 3 archetypes, the tendency of enrichment of the MAP archetype with development of chemo-resistance is already established. Thus this study is at best validation of enrichment of the MAP as well as of PI3K/AKT/MTORC1 in resistant disease, and of the utility of scRNA Seq in longitudinal patient samples to provide data consistent with the already available data.
5. The functional bioenergetics assay relied only on ATP production through OXPhos and glycolysis. The authors should also show ATP production from fatty acid oxidation that has been reported to be unique for HGSC.

Reviewer #2, expert in evolution and archetypal analysis (Remarks to the Author):

I very much enjoyed reading this article which addresses the evolutionary tradeoffs faced by single ovarian cancer cells before and through therapy. The authors propose that evolutionary trade-offs between 3 tasks - metabolism&proliferation, cell defense response, DNA repair - account for much of transcriptional heterogeneity in ovarian cancer.

The importance of these findings changes as cancer develops resistance to therapy, with therapy selecting for metabolism and proliferation and less for cell defense and DNA repair.

Different driver mutations support re-specialization of cells into different tasks.

The work is of high significance for the cancer field because it addresses transcriptional heterogeneity. This transcriptional heterogeneity facilitates tumor progression and the emergence

of resistance to therapy. While descriptions of this heterogeneity have been done in past years, we lack an interpretative framework of the selection factors driving heterogeneity before and through adaptation to treatment. The present study addresses this gap.

Most of the conclusions are supported by the work, and the methods are sound in general. But I have some methodological concerns, especially about the connection with clonality and specialization into tasks which need to be addressed.

Specific points:

1. When introducing the Pareto framework, the authors write "which states that there is a combination of tasks that dominate an organism's fitness (20). The approach defines a polygon, where the number of vertices reflects the number of tasks describing the data" A more precise description should be preferred: "which states that, when a combination of tasks dominate an organism's fitness but the organism cannot be optimal at all tasks at once due to trade-offs, optimal phenotypes should fall on low-dimensional shapes called polyhedra. The number vertices reflects the number of tasks essential to the fitness of the organism."

2. Some small changes could be made to make this manuscript more accessible to scientists the field of ovarian cancer. After all, Nature Communications is a transdisciplinary journal. For example, in the beginning of the results, the authors mention "malignant ascites and pleural effusion", none of which are defined prior in the article. These two terms should be defined what these are for people who don't know this specific cancer type. To this reader, pleural effusion evokes the mucus secreted by lung epithelium- the connection to ovarian cancer is not obvious. This reader also never heard of ascites before.

To fix this, when first introducing HGSOc, explain something like "HGSOc mostly initiates in the epithelium and advances to create compact anatomical substructures called ascites. These ascites then progress to malignancy by invading the mesenchyme and by shedding into the fallop tubes. Following shedding, cancer cells are found in pleural effusion which can be sampled by [name of clinical procedure here]."

3. Statistical validation of the found polyhedra would be appropriate here, which are readily available in the Pareto Task Inference method used by the authors. Mainly, why did the authors choose 3 archetypes, not 2, not 4? To address this, a supplementary figure could show the % of variance explained by 2, 3, 4, ... archetypes, showing that adding a 4th archetype doesn't help explain variance more than 3 archetypes do. At the minimum, the t-ratio test (Hart et al., Nature Methods 2015) should be used to test the statistical significance of fitting a triangle to the data.

4. In the Pareto framework, all tasks contribute to fitness, albeit in different importance depending on the environment. While it's obvious that metabolism&proliferation is fitness enhancing, the authors should briefly discuss how cell defense and DNA repair contribute to the cellular fitness of cancer cells.

5. I find Fig. 7 to be in contradiction with earlier results: the authors previously show that the importance of MAP increases over time, through acquiring resistance to treatment. But they also show that acquiring mutations decreases the importance of MAP in tumors. Because acquiring resistance and acquiring mutations should both follows time, MAP should both increase and decrease with time, a contradiction. To fix this, the authors should comment on this in the discussion

or re-phrase and clarify the paragraphs or legend of Fig. 7.

6. In Fig. 8a and 8b, instead of writing 'cluster', write 'clone', which is more specific and more consistent with the message of the figure.

7. The result that task specialisation is largely clonal is very interesting. But I have an issue with the potentially circular reasoning that supports this claim. Circular reasoning could come about because the authors use gene expression to determine both the position of cells on the Pareto front and the clone through InferCNV. Because of this, the only possible result of this analysis is that clones and task specialization are associated.

To resolve this, another source of evidence than gene expression should be used to determine the clones. One option could be to use mutational signatures from scRNAseq data, if feasible given sequencing depth and seq error rate. There are methods for this, see

<https://genomebiology.biomedcentral.com/articles/10.1186/s13059-019-1863-4>

Alternatively, the authors should provide a convincing argument that the position of single cancer cells on gene expression space is not necessarily determined by the clone as inferred by InferCNV.

Reviewer #3, expert in single cell sequencing (Remarks to the Author):

The manuscript by Nath et al describes a longitudinal scRNA-seq and WGS analysis of ovarian cancer, using ascites and pleural effusion samples. The goal is to understand the diversity of cellular states, the changes in these states over time and with respect to response and resistance to treatments, and the relationship between these changes and genetic diversity. To analyze the single cell data, the authors rely almost exclusively on a specific framework that is driven by an evolutionary perspective, which defines archetype cellular states and describes the diversity as a function of these archetypes. Three such archetypes are defined, relating to metabolism and proliferation (MAP), cellular defense response (CDR) and DNA damage repair (DDR). With some exceptions, the archetypes generally do not show consistent associations with genetics or with development of drug resistance, and hence the analysis does not provide clarity to the origin or consequences of the archetypes.

The main strength of this work is the unique dataset that has been generated. Longitudinal single cell analysis of cancer is currently a major goal in the field and the dataset produced in this work is very impressive. The limitation of the work is that despite the impressive dataset, the results are limited and do not appear to provide an important conceptual advance or to change our understanding of tumor heterogeneity, drug resistance or tumor progression. The analysis is somewhat shallow, focusing with results that are difficult to interpret, and ultimately it is not clear what is the take home message.

Specific comments:

1. The analysis is almost exclusively based on a single computational framework that defines three archetypes and proceeds with the analysis entirely through the lens of these archetypes. This framework is valid and has been used previously, although it might be more specifically suited for other contexts than to cancer cells; this is because in these cancer cells many of the changes reflect immediate responses to the environment as well as genetic drift, which might not fit a framework

that ascribes all effects to adaptation and division of labor between cells. Regardless, the restricted focus on these archetypes feels like an oversimplification that may hide additional aspects of the cellular diversity and hinder a more complete description and understanding of the heterogeneity. It is difficult to fully appreciate the cellular meaning of the archetypes, the degree to which they capture the patterns of cellular diversity and their consistency between patients and time points. I would suspect that more detailed analysis through orthogonal approaches might provide a better description of cellular diversity and perhaps additional results. Below are several suggestions for such additional analysis.

(i) Which genes are correlated (positively or negatively) with progression, as defined by distinct time points, in each individual patient? and what are the overlaps of such genes between patients? This would seem to me like the first step in an unbiased analysis aimed at relating the dataset to tumor progression, which is currently lacking.

(ii) When traditional approaches are used to cluster the malignant epithelial cells in each patient, what are the resulting clusters and their differential expression? How consistent are these patterns of clusters with the archetype analysis and do they uncover additional aspects of cellular diversity?

(iii) When a single archetype is associated with multiple processes (such as proliferation and metabolism in the case of the MAP archetype) are these processes coupled throughout the dataset, or could they be separated in some cases such that MAP values in some cases reflect metabolism and in others reflect cell cycle? In the case of metabolism, could glycolysis, TCA, ox.phos and possibly other aspects be distinguished and compared over patients and time points? In general, the view that these processes are entirely coupled seems like an oversimplification and should be examined further to show the degree of coupling.

(iv) When archetypes reflect a particular process, which genes reflect that process in that archetype, do the relevant genes differ between patients and time points, and what are the exact patterns of those functionally-relevant sets of genes (as opposed to the entire archetype expression profile)? For example, in the case of proliferation, could the current vague definition (i.e. cells belonging to an archetype with multiple aspects including proliferation) be replaced by a concrete classification of single cells into non-dividing and dividing cells, and possibly further classified into phases of the cell cycle? In the case of cellular defense responses, could interferon response be distinguished from other aspects of that archetype?

(v) In terms of visualization, heatmaps that show the archetype-associated genes across single cells would be helpful for evaluating the primary data, the consistency between those distinct genes, and their differences between time-points and patients.

2. The abstract suggests a significant role of MAP in development of resistance, with the statements “The metabolism and proliferation archetype evolved during treatment...” and “...consistent enrichment of subclones with the metabolism archetype as resistance is acquired”. Accordingly, Figure 4 argues for a progression-related increase in the proportion of specialists for the MAP archetype. However, this result seems inconsistent with the main dataset. Specifically, in Figure 8, there is only one patient (7) in which there appears to be a robust longitudinal increase in the MAP archetype, while most patients show other patterns; these include no substantial changes in MAP proportions (patients 5 and 9), changes that are not consistent with time, for example with highest MAP proportion in time point 2 compared to 1 and 3 (patients 6 and 8), and an opposite pattern (patient 4). Accordingly, the data may support an association of MAP with tumor burden (as reflected by CA-125) rather than with tumor progression and drug resistance. This distinction seems to explain the discrepancy between the two analyses (figure 4 and 8) as CA-125 is not always highest in the last time point, while drug resistance is thought to gradually increase and would be expected to be highest at the last time point. An association of metabolism and proliferation expression

profiles with tumor burden seems quite expected, likely reflecting the fact that an increased tumor burden implies more dividing cells and increased energetic demands. Thus, a likely explanation is that increased tumor burden by definition implies an increase in MAP proportion and that MAP is only reflective of proliferating cells and not of a unique cellular state that is being selected for by specific drug treatments.

3. EMT is noted in the discussion as part of the MAP archetype, but this is the first time that EMT is mentioned in the text of the paper (as opposed to the figures). Given the lack of description of EMT patterns in the results, the discussion about EMT seems as if it is coming out of nowhere. It is not clear which EMT-related genes are part of the MAP archetype, how strong is the EMT-related signal, how does it vary between samples and time points and whether it correlates with the proliferation and metabolism aspects of the MAP archetype.

4. Inference of CNAs:

First, in the Methods it is stated that malignant cells were separated from normal cells by CNAs, but this should be shown with a figure that shows CNA profiles of both malignant and normal cells. Second, the separation into distinct subclones of malignant cells by CNA inference, as shown in figure S12, does not seem to be robust, and the exact method by which subclones are defined is not specified. Even if there is variability in CNA patterns it does not seem trivial to define the exact subclonal structure and there is no explanation for how this is done. The patterns of CNAs and the degree to which they support a specific definition of subclones should be better defined and demonstrated.

5. Most claims are made without a proper testing of significance. Statistical tests and associated p-values should be included for all claims.

6. Figures 6 and 7 are not very informative in their current version and might fit better in the supplement.

7. The standard HGSOV abbreviation (which is defined at the top) is then replaced by many instances of HGSOV.

8. The color schemes used in several figures makes it hard to differentiate between patients and cell types and should be replaced.

We sincerely thank the reviewers for their thorough assessment and insightful suggestions for improving our work. We have performed additional experiments and analyses as requested and have addressed the reviewers' concerns in the point-by-point responses below.

REVIEWER COMMENTS

Reviewer #1, expert in high-grade serous ovarian cancer, metabolism and drug resistance (Remarks to the Author):

The authors sought to determine the key phenotypic transcriptomic changes that evolve during chemotherapy due to selective pressure. They employed the Pareto optimization concept, to determine the tasks that dominate an organism's fitness and identified a number and features of driver phenotypes "archetypal phenotypes" in cancer cells that evolve during tumor progression and therapy. The authors also evaluated the archetypes with genetic alterations to identify the potential link between somatic alterations and phenotypic state. The authors identifies 3 archetypes that characterize cancer cell phenotypes during cancer progression and chemotherapy. These included metabolism and proliferation (MAP), cellular defense response (CDR), and DNA damage repair (DDR). The manuscript is well written with comprehensive description of the methodology and analysis. However, I have the following concerns

We appreciate the reviewer's helpful comments and suggestions. We have revised information on the key pathways associated with archetypal phenotypes and performed additional experiments to demonstrate the contribution of fatty acid oxidation to the metabolic shifts.

1. The sample size, though longitudinal over years is small and some patient variabilities cannot be generalized or linked to as specific somatic or germ-line mutation. In addition, the prognostic power of the findings to predict disease outcome (responders or resistance) to a given treatment, or development or recurrence is not addressed.

We agree with the reviewer that the current study is not designed to develop a biomarker and is underpowered to link specific mutations with the observed longitudinal phenotypes or to determine prognostic implications. Due to the nature of this cohort, we refrained from claiming specific mutations (germline or somatic) were associated with HGSOC subclone and archetypes. Of note, while not large in the number of patients, this is the first time that serial samples collected during long-term therapy have been analyzed.

Despite the small sample size, we did observe an interesting correlation between MAP and the overall survival of the patients (see figure below). We found a negative correlation ($r = -0.69$) between the proportion of the MAP archetype specialists (relative to the proportion of all specialists) at the final time point and the overall survival of the patients. This observation suggests that understanding the emergence and the biology of the MAP archetype is relevant for HGSOC progression.

Scatterplot showing a correlation between the relative proportion of MAP specialists at the last time point (Y-axis) vs. overall survival of the patient in days since primary surgery was performed.

2. The authors stated that the genes included in the second archetype cell defense response (CDR) include PI3K/AKT and MTORC1. These signatures are better included with Metabolism and proliferation (MAP) archetype as they represent the main pathways linking cell proliferation to metabolic programming. The authors should revisit their statements and re-assign these pathways.

We appreciate the reviewer's concerns with the assignment of pathways such as PI3K/AKT/MTORC1 to the CDR archetype. We have revised our statements to reflect that the CDR archetype was defined based on the interferon-gamma and IL6/JAK/STAT3 signaling pathways, as was originally shown in Figure 2b.

We have also performed additional analyses to demonstrate the contribution of individual signaling pathways to each archetype. As the classification of the archetypes using the lasso approach shrinks the coefficients of several pathways to zero, we also evaluated the contribution of individual signaling pathways to each archetype using regression analysis. We analyzed the association between KEGG pathway enrichment scores in single cells against the archetype scores of three archetypes across all cells (shown in Supplementary Table 5). The classification of the MAP archetype was supported by the positive associations with enrichment (positive coefficients and $FDR < 0.05$) of pathways including cell cycle, DNA replication, glycolysis, and OXPHOS. The CDR archetype was supported by positive associations with key immune response pathways, including NK cell-mediated cytotoxicity, TLR receptor signaling, RIG-I like receptor signaling, NOD-1-like receptor signaling, and T-cell/B-cell receptor signaling. Interestingly, the CDR archetype scores were also positive correlated with metabolism pathways of glycolysis and OXPHOS, but not with proliferation as indicated by cell cycle and DNA replication, suggesting a decoupling of these two phenotypes in the CDR cells and supporting the separate classification of this cell state from MAP. Furthermore, similar to the classification of the PI3K/MTOR and WNT pathways in the CDR archetype in the group lasso analysis, the CDR cells were again associated with the KEGG MAPK and WNT signaling pathways, suggesting activation of these pathways contributed to the CDR archetype over MAP.

We have added this new information to the section on the classification of archetypes (see Page 10).

3. In lines 196-198, there is discrepancy of patients/ samples numbers of the validation cohort.

Thank you for pointing out the confusion created by the description in lines 196-198. The validation cohort consisted of "unmatched" patient samples obtained from 8 pre-treatment and 7 post-treatment patients. We have modified the sentence to include this information.

4. While the data from the scRNA of longitudinal samples in the pilot and validation cohorts identified malignant cell tendency to cluster into 3 archetypes, the tendency of enrichment of the MAP archetype with development of chemo-resistance is already established. Thus this study is at best validation of enrichment of the MAP as well as of PI3K/AKT/MTORC1 in resistant disease, and of the utility of scRNA Seq in longitudinal patient samples to provide data consistent with the already available data.

We appreciate the reviewer's comment. We have now performed additional analyses with the scRNA-seq data to provide a more detailed insight into the phenotypes that encompass the archetypes. As we have discussed in response to the reviewer's comment #2, we found that the MAP and CDR archetypes are inherently decoupled, despite sharing key metabolic phenotypes. We have also found that MAPK or WNT signaling pathways are associated with the CDR archetype, rather than the MAP cells, suggesting their contributions to cells impacted by immune surveillance over cells undergoing active proliferation. At the last time point, we observed a significant negative correlation between the CDR and MAP cells (Supplementary Figure 15d). Thus, we observed an overall shift from CDR to MAP specialists at later time points along the course of progression, while the shift towards MAP specialists at the last time point indicated a link with poor survival. Together with the association of archetypal cell populations with specific subclones, our results suggest that the MAP cells are a distinct population of cells that should be further evaluated to identify their origins and to develop strategies to combat their evolution.

5. The functional bioenergetics assay relied only on ATP production through OXPhos and glycolysis. The authors should also show ATP production from fatty acid oxidation that has been reported to be unique for HGSC.

As suggested by the reviewer, we have performed additional assays to characterize the contribution of fatty acid oxidation to the ATP production in the HGSOC cells over time. We determined the metabolic capacity of the cell lines derived from patients 4 and 8 at the early and late time points in the presence or absence of a fatty acid oxidation (FAO) inhibitor. Our results indicate that the inhibition of FAO resulted in a significant decrease in the total ATP production of the cells. However, this reduction was consistent over time, suggesting the contribution of FAO did not change over time (see figure below).

These results were added to the results section on Page 15:

“We also evaluated the contribution of fatty acid oxidation by comparing the contribution to ATP production in the presence of a fatty acid oxidation inhibitor. We found a significant decrease in ATP production, but this reduction was consistent over time (Supplementary Figure 16).”

Supplementary Figure 16. Barplots showing the mean and SEM ($n = 3$) of percentage total oxygen consumption rate in cell lines treated with fatty acid oxidation inhibitors. The left panel shows data from cell lines from patient 4 derived at the first and last time points, and the right panel shows data from cell lines from patient 8 at three time points. The horizontal bars above the plot show pairwise comparisons and results of t-tests ($n.s.$ = not significant, $p > 0.05$).

Reviewer #2, expert in evolution and archetypal analysis (Remarks to the Author):

I very much enjoyed reading this article which addresses the evolutionary tradeoffs faced by single ovarian cancer cells before and through therapy. The authors propose that evolutionary trade-offs between 3 tasks - metabolism&proliferation, cell defense response, DNA repair - account for much of transcriptional heterogeneity in ovarian cancer. The importance of these findings changes as cancer develops resistance to therapy, with therapy selecting for metabolism and proliferation and less for cell defense and DNA repair. Different driver mutations support re-specialization of cells into different tasks. The work is of high significance for the cancer field because it addresses transcriptional heterogeneity. This transcriptional heterogeneity facilitates tumor progression and the emergence of resistance to therapy. While descriptions of this heterogeneity have been done in past years, we lack an interpretative framework of the selection factors driving heterogeneity before and through adaptation to treatment. The present study addresses this gap. Most of the conclusions are supported by the work, and the methods are sound in general. But I have some methodological concerns, especially about the connection with clonality and specialization into tasks which need to be addressed.

We thank the reviewer for their time and insight while reviewing this research, as well as for their encouraging comments and for pointing out the methodological concerns. In particular, we are grateful for the advice on how to most effectively communicate our findings. We have addressed these concerns by providing the relevant analyses that support the selection of number of archetypes and details of the subclonal analyses.

Specific points:

1. When introducing the Pareto framework, the authors write "which states that there is a combination of tasks that dominate an organism's fitness (20). The approach defines a polygon, where the number of vertices reflects the number of tasks describing the data" A more precise description should be preferred: "which states that, when a combination of tasks dominate an organism's fitness but the organism cannot be optimal at all tasks at once due to trade-offs, optimal phenotypes should fall on low-dimensional shapes called polyhedra. The number vertices reflects the number of tasks essential to the fitness of the organism."

We sincerely appreciate this suggestion and have modified the sentence as recommended by the reviewer in the introduction on Page 5.

2. Some small changes could be made to make this manuscript more accessible to scientists the field of ovarian cancer. After all, Nature Communications is a transdisciplinary journal. For example, in the beginning of the results, the authors mention "malignant ascites and pleural effusion", none of which are defined prior in the article. These two terms should be defined what these are for people who don't know this specific cancer type. To this reader, pleural effusion evokes the mucus secreted by lung epithelium- the connection to ovarian cancer is not obvious. This reader also never heard of ascites before. To fix this, when first introducing HGSOC, explain something like "HGSOC mostly initiates in the epithelium and advances to create compact anatomical substructures called ascites. These ascites then progress to malignancy by invading the mesenchyme and by shedding into the fallop tubes. Following shedding, cancer cells are found in pleural effusion which can be sampled by [name of clinical procedure here]."

As suggested, we have now included a brief description in the introduction on Page 3: "A majority of HGSOCs arise from the epithelium of fallopian tubes (8), often resulting in detection of malignant cells that escape into the fluids accumulating in the peritoneal cavity (ascites) or in the lung pleural effusions following late-stage extra-abdominal metastases (9)."

3. Statistical validation of the found polyhedra would be appropriate here, which are readily available in the Pareto Task Inference method used by the authors. Mainly, why did the authors choose 3 archetypes, not 2, not 4? To address this, a supplementary figure could show the % of variance explained by 2, 3, 4, ... archetypes, showing that

adding a 4th archetype doesn't help explain variance more than 3 archetypes do. At the minimum, the t-ratio test (Hart et al., Nature Methods 2015) should be used to test the statistical significance of fitting a triangle to the data.

We agree with the reviewer that the choice of selecting 3 archetypes should be explained for each archetype analysis performed in this study. As suggested, we have provided the variance explained by varying number of archetypes from 3-8 in all the analyses we have performed. We have also provided the gain in variance explained by the model with increment in the number of archetypes from 3-8. Specifically, we have provided this information for the integrated longitudinal cohort data in Supplementary Figure 4, integrated validation cohort data in Supplementary Figure 10 and individual longitudinal cohort data in Supplementary Figure 12.

We have also added the following description in the main text in the results section on Pages 8-9: “To determine the shape of a polygon that can best enclose the data, we performed simulations with a varying number of vertices ranging from 3 to 8 (Supplementary Figure 4). There was a minimal gain in the variance explained by the models with >3 archetypes, showing that a triangle was enough to enclose the data. Moreover, any gain in variance explained by the models with >3 archetypes was at the cost of increased uncertainty in the position of vertices, and a decrease in the ratio of the volume of the polytope to the convex hull (t-ratio) confirmed that the 3-vertex triangle reliably enclosed the data (Figure 2a).”

Supplementary Figure 4. Simulations with 3-8 archetypes comparing variance explained by each number of archetypes (left panel), variance in position of the archetypes (middle panel) and the ratio of volume of the polytope to the convex hull or t-ratio (right panel). The results show a minimal gain in variance explained upon increasing the number of archetypes, at the cost of increased variance in the position of the archetypes and a reduction in t-ratio.

4. In the Pareto framework, all tasks contribute to fitness, albeit in different importance depending on the environment. While it's obvious that metabolism & proliferation is fitness enhancing, the authors should briefly discuss how cell defense and DNA repair contribute to the cellular fitness of cancer cells.

Thank you for this important suggestion. Based on the reviewer's suggestion and additional results from new analyses, we have updated the discussion section to include the following information on Pages 21-22: “We found that the metabolism and proliferation archetype (MAP) evolved later over the course of chemotherapy compared to early time points or treatment-naïve samples and, at the last time point, this was concomitant with a decrease in the CDR archetype and correlated with poor overall survival of the patients (Supplementary Figure 15). Our results support the clinical observation that exceptional long-term HGSOC survivors are associated with enrichment of immune response signatures while short-term survivors tend to be associated with proliferation signatures (39).

Interestingly, pathways that are well known to contribute to cellular survival, metabolism, and proliferation from HGSOC bulk transcriptomes, such as MAPK (40) and WNT (41) signaling, were preferentially associated with non-proliferating cells in the CDR archetype, instead of the proliferating cells of the MAP archetype. The pathways enriched in the CDR archetype like TLR signaling and NK-cells mediated cytotoxicity suggest that these cells are responsive to immune cells in the microenvironment (42). Thus, the prevalence of CDR cells could be indicative of active immune surveillance in the tumor, which has been linked to better prognosis and outcomes of ovarian cancers (43). Key metabolic pathways (OXPHOS, glycolysis) were also associated with the CDR archetype (Supplementary Table 5). Based on this, it is reasonable that a chemoresistant tumor would select metabolically active MAP cells that are actively proliferating, instead of metabolically active CDR cells that are subject to immune surveillance. This observation also supports the idea of multi-task evolution, where the progressive tumors select for cells specializing in proliferation over immune response, assuming both cell states have similar fitness costs as indicated by enrichment of metabolic pathways.”

5. I find Fig. 7 to be in contradiction with earlier results: the authors previously show that the importance of MAP increases over time, through acquiring resistance to treatment. But they also show that acquiring mutations decreases the importance of MAP in tumors. Because acquiring resistance and acquiring mutations should both follow time, MAP should both increase and decrease with time, a contradiction. To fix this, the authors should comment on this in the discussion or re-phrase and clarify the paragraphs or legend of Fig. 7.

We understand the confusion arising from Figure 7 and description of the results. We have evaluated the association between the presence of key somatic mutations in tumor samples across all time points and the likelihood that a particular archetype might be enriched in a tumor carrying the somatic mutation. As we have shown in Figure 7, most of the common mutations are not associated with a particular archetype. That said, it is very much possible that the emergence of the archetypes can be explained other somatic mutations or epigenetic changes that are not yet characterized to be associated with HGSOC progression. Unfortunately, our study is not statistically powered to identify new somatic mutations associated with progression. Thus, the data does not contradict the fact somatic variants are acquired over time and only highlights that known driver somatic variants are not associated with the emergence of archetypes.

We have clarified this result in the text and updated the discussion section to reflect this information on Page 24: “We found that driver somatic mutations were not associated with the emergence of archetypes across the patients. We have evaluated the association between the presence of key somatic mutations in tumor samples across all time points and the likelihood that a particular archetype might be enriched in a tumor carrying the somatic mutation (Figure 7). While most of the common mutations are not associated with a particular archetype it is possible that the emergence of the archetypes can be explained by other somatic mutations changes that are not yet characterized to be associated with HGSOC progression. As our study is underpowered to discover new somatic variants, our results do not completely rule out the potential role of genetic mechanisms in archetypal evolution, as evidenced by the close association of archetype shifts with specific subclones.”

6. In Fig. 8a and 8b, instead of writing 'cluster', write 'clone', which is more specific and more consistent with the message of the figure.

As suggested, we have updated Figure 8 to replace “cluster” with “subclone”.

7. The result that task specialisation is largely clonal is very interesting. But I have an issue with the potentially circular reasoning that supports this claim. Circular reasoning could come about because the authors use gene expression to determine both the position of cells on the Pareto front and the clone through InferCNV. Because of this, the only possible result of this analysis is that clones and task specialization are associated.

To resolve this, another source of evidence than gene expression should be used to determine the clones. One option could be to use mutational signatures from scRNAseq data, if feasible given sequencing depth and seq error rate. There are methods for this, see <https://genomebiology.biomedcentral.com/articles/10.1186/s13059-019->

1863-4

Alternatively, the authors should provide a convincing argument that the position of single cancer cells on gene expression space is not necessarily determined by the clone as inferred by InferCNV.

We understand the reviewers concern regarding the analyses comparing scRNA-seq derived archetypes with subclonal population called using the InferCNV method which also relies on scRNA-seq data. While we agree that alternative approaches that utilize variant information could help support the subclonal structure, we have found that the 10X scRNA-seq data was sparse and not suitable for clustering-based resolution of subclonal structures using somatic mutations. Nevertheless, we believe that InferCNV is a suitable approach because it estimates the regions of gain and loss by applying a moving average over a sliding window across genes, sorted by their chromosomal locations. Further, the default method reduces the expression values of outliers to a set threshold that minimize the impact of single genes on the moving average. Thus, the inferred regions of gain or loss and the resulting clusters are more likely to reflect genetic subclones based on regains of gain or loss than transcriptional clusters.

As suggested by the reviewer, we have also provided additional details for our inference of the subclonal structure of the HGSOC tumors using the InferCNV method. Briefly, our approach to define the subclones utilized the InferCNV subclusters determined by a hidden markov model (HMM), followed by inspection of regions of copy number gain or loss in each predicted subcluster. In case of ambiguous regions of gain or loss, we also assessed the presence of these regions in our WGS data. Finally, we determined the branches of the clonal evolutionary tree based on the patterns of shared regions of gain or loss. We have provided a detailed description of the procedure and associated figures in a new supplement to the methods. As shown in the CNA figures for each patient, the subclusters/subclones are clearly defined by regional copy number alterations. Please see “Supplementary Methods” for a detailed description and figures showing CNA’s that were used to determine the subclones.

Reviewer #3, expert in single cell sequencing (Remarks to the Author):

The manuscript by Nath et al describes a longitudinal scRNA-seq and WGS analysis of ovarian cancer, using ascites and pleural effusion samples. The goal is to understand the diversity of cellular states, the changes in these states over time and with respect to response and resistance to treatments, and the relationship between these changes and genetic diversity. To analyze the single cell data, the authors rely almost exclusively on a specific framework that is driven by an evolutionary perspective, which defines archetype cellular states and describes the diversity as a function of these archetypes. Three such archetypes are defined, relating to metabolism and proliferation (MAP), cellular defense response (CDR) and DNA damage repair (DDR). With some exceptions, the archetypes generally do not show consistent associations with genetics or with development of drug resistance, and hence the analysis does not provide clarity to the origin or consequences of the archetypes.

The main strength of this work is the unique dataset that has been generated. Longitudinal single cell analysis of cancer is currently a major goal in the field and the dataset produced in this work is very impressive. The limitation of the work is that despite the impressive dataset, the results are limited and do not appear to provide an important conceptual advance or to change our understanding of tumor heterogeneity, drug resistance or tumor progression. The analysis is somewhat shallow, focusing with results that are difficult to interpret, and ultimately it is not clear what is the take home message.

We thank the reviewer for their critical insight and for taking the time to provide a thorough assessment of our work. Specifically, we appreciate the advice regarding additional orthogonal analyses, which have provided new insights into our data and helped mitigate the concerns raised by the reviewer.

Specific comments:

1. The analysis is almost exclusively based on a single computational framework that defines three archetypes and proceeds with the analysis entirely through the lens of these archetypes. This framework is valid and has been used previously, although it might be more specifically suited for other contexts than to cancer cells; this is because in these cancer cells many of the changes reflect immediate responses to the environment as well as genetic drift, which might not fit a framework that ascribes all effects to adaptation and division of labor between cells. Regardless, the restricted focus on these archetypes feels like an oversimplification that may hide additional aspects of the cellular diversity and hinder a more complete description and understanding of the heterogeneity. It is difficult to fully appreciate the cellular meaning of the archetypes, the degree to which they capture the patterns of cellular diversity and their consistency between patients and time points. I would suspect that more detailed analysis through orthogonal approaches might provide a better description of cellular diversity and perhaps additional results. Below are several suggestions for such additional analysis.

(i) Which genes are correlated (positively or negatively) with progression, as defined by distinct time points, in each individual patient? and what are the overlaps of such genes between patients? This would seem to me like the first step in an unbiased analysis aimed at relating the dataset to tumor progression, which is currently lacking.

(ii) When traditional approaches are used to cluster the malignant epithelial cells in each patient, what are the resulting clusters and their differential expression? How consistent are these patterns of clusters with the archetype analysis and do they uncover additional aspects of cellular diversity?

(iii) When a single archetype is associated with multiple processes (such as proliferation and metabolism in the case of the MAP archetype) are these processes coupled throughout the dataset, or could they be separated in some cases such that MAP values in some cases reflect metabolism and in others reflect cell cycle? In the case of metabolism, could glycolysis, TCA, ox.phos and possibly other aspects be distinguished and compared over patients and time points? In general, the view that these processes are entirely coupled seems like an oversimplification and should be examined further to show the degree of coupling.

(iv) When archetypes reflect a particular process, which genes reflect that process in that archetype, do the relevant genes differ between patients and time points, and what are the exact patterns of those functionally-

relevant sets of genes (as opposed to the entire archetype expression profile)? For example, in the case of proliferation, could the current vague definition (i.e. cells belonging to an archetype with multiple aspects including proliferation) be replaced by a concrete classification of single cells into non-dividing and dividing cells, and possibly further classified into phases of the cell cycle? In the case of cellular defense responses, could interferon response be distinguished from other aspects of that archetype?

(v) In terms of visualization, heatmaps that show the archetype-associated genes across single cells would be helpful for evaluating the primary data, the consistency between those distinct genes, and their differences between time-points and patients.

We agree with the reviewer's comment that additional analyses can augment the results of the archetype analyses and appreciate the thoughtful suggestions. We believe that the reviewer raised an important concern, that the changes reflect immediate changes in the environment and genetic drift. We have shown in our analyses that key somatic variants acquired over time (Figure 6) are not associated with the emergence of archetypes (Figure 7). Additionally, we have found that the archetypes and the contributing pathways are consistent across patients and over time supported by the new analyses detailed below.

Once again, we appreciate the reviewer's advice and have performed several new analyses to further explore the genes and pathways that are differentially expressed over time, the relationship between transcriptional clusters and archetypes, and the relationship between the pathways that distinguish individual clusters and archetypes.

1. We evaluated the temporal single cell expression patterns of the genes across all patients using a regression analysis of the batch-corrected genes from the 10X scRNA-seq profiles of the longitudinal cohort patients against time. We selected the top 100 upregulated and 100 downregulated genes ranked by FDR to study the overlap of temporally differentially expressed genes shared between patients (Supplementary Figure 3). Very few differentially expressed genes were commonly shared across the patients. Only 3 upregulated genes were shared by more than three patients, including *LCN2* (4 patients), *KRT18* (3 patients) and *SAA1* (3 patients). None of the downregulated were shared by more than three patients. Given these observations, we adapted a pathway-centric approach to focus on the evolution of longitudinal phenotypes in the HGSOC single-cells instead of sparse individual genes.

Supplementary Figure 3. Upset plots displaying the frequency of unique and overlapping differentially expressed genes between the patients. The top panel shows the overlap of the top 100 upregulated genes over time while the bottom panel shows top 100 down regulated genes.

2. As the classification of the archetypes using the lasso approach shrinks coefficients of several pathways to zero, we also evaluated the contribution of individual signaling pathways to each archetype using regression analysis. We analyzed the association between KEGG pathway enrichment scores in single cells against the archetype scores of three archetypes across all cells (Supplementary Table 5). The classification of the MAP archetype was supported by the positive associations with enrichment (positive coefficients and $FDR < 0.05$) of key pathways including cell cycle, DNA replication, glycolysis and OXPHOS. The CDR archetype was supported by positive associations with key immune response pathways, including NK cells-mediated cytotoxicity, TLR receptor signaling, RIG-I like receptor signaling, NOD-I-like receptor signaling and T-cell/B-cell receptor signaling. Interestingly, the CDR archetype scores were also positive correlated with metabolism pathways of glycolysis and OXPHOS, but not with proliferation as indicated by cell cycle and DNA replication, suggesting a decoupling of these two phenotypes in the CDR cells and supporting separate classification of this cell state from MAP. Furthermore, similar to the classification of the PI3K/MTOR and WNT pathways in the CDR archetype in the group lasso analysis, the CDR cells were again associated with the KEGG MAPK and WNT signaling pathways, suggesting activation of these pathways contributed to the CDR archetype over MAP.
3. Across the single cells, the key pathways contributing to the MAP archetype (Supplementary Figure 7A) were positively intercorrelated (Pearson's correlation coefficient > 0 , $FDR < 0.05$) (Supplementary Figure 7B), with subtle differences observed between subpopulations of cells classified based on cell cycle states

(Supplementary Figure 7C) and between patients across time (Supplementary Figure 7D and 7E). For example, the MAP cells were more metabolically active in S and G1 cells compared to G2M cells, as expected. The MAP phenotypes on an average were largely consistent overtime, with some significant shifts observed within specific patients, like decreased glycolysis in patient 5, and increased glycolysis and OXPHOS in patient 8. The key pathways contributing to the CDR pathways were also highly intercorrelated (Pearson's correlation coefficient > 0, FDR < 0.05) (Supplementary Figure 8A), with some differences observed in specific patients over time (Supplementary Figure 8B). In particular, patients 5 and 9 showed reduced enrichment of multiple immune response pathways over time.

Supplementary Figure 7A. Heatmap displaying the gene set enrichment scores of key metabolism and proliferation KEGG pathways across single cells classified as MAP specialists. B. Correlation plot of key metabolism and proliferation KEGG pathways. The colors indicate magnitude of Pearson's correlation between the enrichment scores. C-D Ridge plots showing distribution of pathway enrichment scores across phases of cell cycle (C) or time (D). The vertical bars annotated with * show pairwise comparisons that are statistically significant from TukeyHSD test following ANOVA. E. Violin plots showing comparison of pathway enrichment scores across time in each patient. * indicates statistically significant difference between time 3 and time 1 in the pairwise comparison from TukeyHSD test following ANOVA. A $P < 0.05$ is considered statistically significant.

Supplementary Figure 8A. Correlation plot of key immune response related KEGG pathways across single cells classified as CDR specialists. The colors indicate magnitude of Pearson's correlation between the enrichment scores. B. Violin plots showing comparison of pathway enrichment scores across time in each patient. * indicates statistically significant difference between time 3 and time 1 in the pairwise comparison from TukeyHSD test following ANOVA. A $P < 0.05$ is considered statistically significant.

- Next, we compared the distribution of the archetype specialists across single-cell transcriptional clusters of malignant cells and their distribution over time (Supplementary Figure 15A). The CDR specialists were largely associated with malignant cell cluster 0, while DDR specialists were associated with cluster 1 and MAP specialists with cluster 2 (Supplementary Figure 15B). Clusters 4 and 5 were small clusters with few cells, which were present transiently at time 1 or time 2. At the first time point, cluster 2 and MAP specialists were present at the lowest proportions compared to other archetypes, indicating that the cluster 2/MAP specialists were acquired along the course of progression (Supplementary Figure 15B). The proportion of specialists at the last time point relative to other archetypes were not the same across all the patients. Specifically, the proportion of the MAP and CDR archetypes showed strong negative correlation (Supplementary Figure 15C). This relative shift between the archetypes impacted the overall survival of the patients (time to death in days since primary surgery was performed). While this sample size is very small, we found a negative correlation between the proportion of MAP specialists in a patient and overall survival (Pearson's correlation = -0.69, $R^2 = 0.48$) (Supplementary Figure 15D). Thus, we observed an overall tendency of MAP specialists to evolve at later time points along the course of

progression, while the shift towards MAP specialists at the last time point indicated a link with poor survival.

Supplementary Figure 15A. UMAP projections of batch-corrected cancer cells colored by patient (top left), time (top right), Seurat clusters (bottom left) or archetype specialist type (bottom right). B. Barplots showing absolute number of archetype specialists in different Seurat clusters (left panel), with stacked barplots (middle and right panels) showing percentage of cells in each Seurat cluster or specialist cluster grouped by time. C-D Scatterplots comparing the relative portion of MAP specialists at the last time point with the relative proportion of CDR specialists at the last time point (C) or overall survival in days since primary surgery was performed (D).

2. The abstract suggests a significant role of MAP in development of resistance, with the statements “The metabolism and proliferation archetype evolved during treatment...” and “...consistent enrichment of subclones with the metabolism archetype as resistance is acquired”. Accordingly, Figure 4 argues for a progression-related increase in the proportion of specialists for the MAP archetype. However, this result seems inconsistent with the main dataset. Specifically, in Figure 8, there is only one patient (7) in which there appears to be a robust longitudinal increase in the MAP archetype, while most patients show other patterns; these include no substantial changes in MAP proportions (patients 5 and 9), changes that are not consistent with time, for example with highest MAP proportion in time point 2 compared to 1 and 3 (patients 6 and 8), and an opposite pattern (patient 4). Accordingly, the data may support an association of MAP with tumor burden (as reflected by CA-125) rather than with tumor progression and drug resistance. This distinction seems to explain the discrepancy between the two analyses (figure 4 and 8) as CA-125 is not always highest in the last time point, while drug resistance is thought to gradually increase and would be expected to be highest at the last time point. An association of metabolism and proliferation expression profiles with tumor burden seems quite expected, likely reflecting the fact that an increased tumor burden implies more dividing cells and increased energetic demands. Thus, a likely explanation is that increased tumor burden by definition implies an increase in MAP proportion and that MAP is only reflective of proliferating cells and not of a unique cellular state that is being selected for by specific drug treatments.

We agree with the reviewer’s assessment that the emergence of the MAP proportions is likely indicative of tumor burden by definition. The CA-125 level at each time points of our study overlapping with sample collection for scRNA-seq were consistently much greater than 100 U/ml (Figure 1, Supplementary Table 3), suggesting the tumor

burden indicative of poor prognosis was already very high at all time points, and also not necessarily correlated with overall survival of the patients (Supplementary Table 1). We appreciate this insightful comment and, given this discrepancy, we have revised our abstract and removed the discussion of CA-125 levels from the viewpoint of prognosis.

Interestingly, the detailed analyses suggested by the reviewer in comment #1 have revealed a new insight that the metabolism phenotype is in fact associated with both the CDR and MAP archetypes; however, it is decoupled from proliferation in the CDR archetype. Moreover, the emergence of MAP specialists at the last time point is negatively correlated with CDR and appears to be associated with poor survival outcomes. Thus, it will be pertinent to study the difference in the biology of the MAP archetype and their mechanism of selection.

Accordingly, we have added the following results on Page 14:

“The proportion of specialists changed over time in patients. Specifically, the proportion of the MAP and CDR archetypes showed a strong negative correlation (Pearson’s correlation = -0.95, $R^2 = 0.9$) (Supplementary Figure 15C). This relative shift between the archetypes impacted the overall survival of the patients (time to death in days since primary surgery was performed). While this sample size is small, we found a negative correlation between the proportion of MAP specialists in a patient at the final time point and overall survival (Pearson’s correlation = -0.69, $R^2 = 0.48$) (Supplementary Figure 15D). Thus, we observed an overall tendency of MAP specialists to evolve at later time points along the course of progression, while the shift towards MAP specialists at the last time point indicated a link with poor survival.”

We added the following paragraph to the discussion section on Pages 21-22:

“We found that the metabolism and proliferation archetype (MAP) evolved later over the course of chemotherapy compared to early time points or treatment-naïve samples and, at the last time point, this was concomitant with a decrease in the CDR archetype and correlated with poor overall survival of the patients (Supplementary Figure 15). Interestingly, pathways that are well known to contribute to cellular survival, metabolism and proliferation from HGSOC bulk transcriptomes, such as MAPK and WNT signaling, were preferentially associated with non-proliferating cells in the CDR archetype, instead of the proliferating cells of the MAP archetype. The pathways enriched in the CDR archetype, like TLR signaling and NK-cells mediated cytotoxicity suggest that these cells are responsive to immune cells in the microenvironment. Thus, the prevalence of CDR cells could be indicative of active immune surveillance in the tumor, which has been linked to better prognosis and outcomes of ovarian cancers. Key metabolic pathways (OXPHOS, glycolysis) were also associated with the CDR archetype (Supplementary Table 5). Based on this, it is reasonable that a chemoresistant tumor would select metabolically active MAP cells that are actively proliferating, instead of metabolically active CDR cells that are subject to immune surveillance. This observation also supports the idea of multi-task evolution, where the progressive tumors select for cells specializing in proliferation over immune response, assuming both cell states have similar fitness costs as indicated by enrichment of metabolic pathways.”

3. EMT is noted in the discussion as part of the MAP archetype, but this is the first time that EMT is mentioned in the text of the paper (as opposed to the figures). Given the lack of description of EMT patterns in the results, the discussion about EMT seems as if it is coming out of nowhere. It is not clear which EMT-related genes are part of the MAP archetype, how strong is the EMT-related signal, how does it vary between samples and time points and whether it correlates with the proliferation and metabolism aspects of the MAP archetype.

Thank you for this comment. We have updated the results section on Page 9 to indicate that hallmark EMT pathway was one of the key predictors of the MAP archetype, as shown in the group-lasso cross-validation analyses with the hallmark pathways and archetype scores. Given the sparsity of individual genes, we have limited the discussion on the correlation between EMT pathway and the MAP archetype to the discussion section. Specifically, we have added “In addition to the metabolic and proliferation pathways, we also observed a consistent emergence of the epithelial to mesenchymal transition (EMT) pathway as one of the key hallmark predictors of the MAP archetype (Supplementary Figures 5, 6, 11, and 13)” on Page 23.

4. Inference of CNAs:

First, in the Methods it is stated that malignant cells were separated from normal cells by CNAs, but this should be shown with a figure that shows CNA profiles of both malignant and normal cells.

Second, the separation into distinct subclones of malignant cells by CNA inference, as shown in figure S12, does not seem to be robust, and the exact method by which subclones are defined is not specified. Even if there is variability in CNA patterns it does not seem trivial to define the exact subclonal structure and there is no explanation for how this is done. The patterns of CNAs and the degree to which they support a specific definition of subclones should be better defined and demonstrated.

We agree with the reviewer that it is difficult to define the exact subclonal structure of the tumors based on the CNA profiles. As suggested by the reviewer, we have provided a detailed description of the procedure we followed to determine the malignant cells, followed by a description of the method we followed to determine the subclonal structure of each patient. These analyses and results have been added to Supplementary Methods.

“To identify malignant epithelial cells, we used a first-pass classification filter to separate epithelial cells from immune cells and stromal components. Since the samples were obtained from malignant pleural effusions and ascites, it was reasonably expected that the epithelial cells identified in the samples were malignant. These cells appeared in separate clusters in UMAPs and were verified with the expression of key marker genes (Supplementary Figure 2). To further affirm the classification of the epithelial cells, we used the fibroblasts obtained from an HGSOC tumor as reference to perform inferCNV analysis with the epithelial cells from individual HGSOC patients across all time points. The parameters for the inferCNV analysis were: cutoff = 0.1 (as recommended for 10X scRNA-seq data), min_cells_per_gene = 3, with default hidden markov model (HMM) and denoise set to true. The resulting profiles displayed the presence of CNAs in all epithelial cells compared to the normal reference, thus confirming the classification of malignant epithelial cells. The heatmaps on the subsequent pages also display the expression of key epithelial markers, such as *EPCAM*, *KRT8* and *KRT18* across the single cells, along with the absence of immune cell markers like *CD45*, *THY1*, *CD3E* and *CD68*.

To construct the subclonal structure for each patient, we used the subcluster method with the HMM CNA profiles to first determine the number of subclones and distribution of the cells in each subclone. Next, we inspected the HMM and denoised CNA profiles to identify regions of copy number gains or losses that could verify the subclonal structure determined by the HMM subcluster method. We also used the whole genome sequencing profiles in case of patients where the regions of gain or losses that distinguished subclones were difficult to assign using the HMM or CNA profiles alone. After the subclonal structures were confirmed and individual cells assigned to each subclone, we derived the evolutionary tree as follows: First, the proportion of cells assigned to each subclone were used to determine the frequency of subclones at each time point. Then, the branches of the tree were assigned on the basis of shared regions of gain or loss. For example, in the case of Patient 4 (see figure on next page) the three subclones were first determined using the HMM subclusters. Then, Subclone 1 was distinguished from subclones 2 and 3 by the presence of a unique region of gain on chromosomes 1 and 17. Subclones 2 and 3 shared a difference in region of gain on chromosome 17 seen in subclone 2, while subclone 3 was characterized by unique regions of loss on chromosomes 4 and X. These regions are encircled in black. Thus, we determined that subclone 3, which appears in samples collected at the second time point, evolved from subclone 2. Finally, the distribution of the cells in the three subclones were used to define the subclone frequencies at each time point. This procedure was followed for other samples, where the HMM subclusters were first confirmed by the presence of unique regions of gain or loss, followed by assignment of the evolutionary branching based on shared and acquired CNAs, and determination of subclone frequencies.”

5. Most claims are made without a proper testing of significance. Statistical tests and associated p-values should be included for all claims.

Thank you for pointing this error. We have provided results of statistical tests and revised claims accordingly.

Specifically,

1. We have provided regression coefficients, p-value and FDR for pathway associations with archetype scores. This is to support the classification of the archetypes based on group-lasso coefficients from cross-validation analyses that we did not describe using conventional statistics (Supplementary Table 5).
2. We modified the Figure 4b to display boxplots of proportions of the archetype specialists to replace the barplots displaying difference and annotated the plots to show p-values.
3. We modified Figures 4c-d to show individual data points and added the p-values from pairwise t-tests for difference in mean metabolic activities adjusted for multiple comparisons.
4. We annotated the new supplementary figures based on the reviewer's suggestions with statistical tests. For correlations, we provided Pearson's correlation coefficient, R^2 and p-values. For analyses with multiple contrasts, we provided p-values from TukeyHSD posthoc test following ANOVA (Supplementary Figures 7, 8 and 9).
5. The associations between archetypes and subclones were evaluated with TukeyHSD following ANOVA, as indicated (Supplementary Figure 18).

6. Figures 6 and 7 are not very informative in their current version and might fit better in the supplement.

The somatic alterations reported in these analyses are likely of general interest to the ovarian cancer community, where they have been extensively studied for their role in chemoresistance. Thus, we believe this information might be important to share, especially to emphasize that these somatic alterations are not correlated with the emergence of phenotypic archetypes.

7. The standard HGSOC abbreviation (which is defined at the top) is then replaced by many instances of HGSOV.

Thank you for pointing this error. We have made corrections to keep the abbreviation consistent throughout the manuscript.

8. The color schemes used in several figures makes it hard to differentiate between patients and cell types and should be replaced.

As suggested, we have modified several figures to better reflect the key point behind the figure.

1. We modified Figure 1c to show cells clustered on UMAP colored by cell types instead of patient ID. Several additional visualizations of the UMAP projections of the cancer cells are also provided in the new Supplementary Figure 15.
2. We modified Figure 2a to show cells colored by archetype scores, as the coloring by patient ID did not provide any additional information.
3. We modified Figure 2d to show the cell clusters prior to CCA normalization for easy visualization of sample clusters and small populations of normal cells. Further, we modified the color scheme for patients for easy visualization of the early and late cohorts in Figure 2e.

REVIEWER COMMENTS

Reviewer #1 (Remarks to the Author):

The authors adequately addressed my concerns to my satisfaction.

Reviewer #2 (Remarks to the Author):

Reviewer 2: the authors have addressed all of my concerns and my suggestions diligently, except for point 3 of the original rebuttal letter.

The choice of 3 archetypes now looks compelling and is well-supported by Fig S4A.

But I can't say that the authors convinced me that a polytope is a good fit for the data. Showing that a polytope is a good fit for the data statistically-speaking is important because the trade-off interpretation rests on the claim that the data fits a polytope.

Specifically, I have 3 issues with the panel on the t-ratio test in Fig S4B:

1. The authors represent the ratio of the volume of the best-fitting polytope over the volume of the convex hull (CH). Because the CH has more degrees of freedom and because no data point can be outside the polytope, the CH always fits the data better than a polytope. Thus, the volume of the CH is smaller than the volume of the polytope, so that the ratio shown on Fig S4B should be larger than 1. Yet, in Fig S4B, this ratio is always smaller than 1. How can this be?
2. I will assume that there was a labeling mistake in Fig S4B, and that the ratio represented is actually the volume of the CH over the volume of the best-fitting polytope. I will also assume that this ratio in many randomized datasets is represented as a probability density in Fig S4B, together with the t-ratio in the non-randomized data as a vertical line (is this correct? If so, please specify in the legend). There, we see that the t-ratio is increased in randomized data compared to non-randomized data. This implies that a polytope fits the randomized data better than the original, non-randomized data. Such an observation that does not support that a polytope fits the data.
3. Even if I assume that the densities and the line represented in the t-ratio panel of Fig S4B represent other ratios than the one I guessed, the p-values shown do not support the claim that a polytope is a better fit to the data compared to randomized data. The p-value for 3 archetypes is apparently 0.9, which is overwhelming evidence for accepting the null hypothesis (the polytope is a bad fit). A minimal p-value threshold to reject the null hypothesis and thus conclude that the polytope is a good fit would be 0.05. But even this threshold implies too large a type I error so that a threshold of 1% is recommended by statisticians (Ioannidis, PLoS Medicine 2005).

These points need to be addressed showing that a polytope is a statistically significant fit for the data underlies the trade-off interpretation.

Perhaps the easiest way to do so is to use the ParTI software (Hart et al. Nature Methods 2015) with 3 archetypes in order to compute the p-value of t-ratio test. Showing a significant p-value there ($p < 0.05$ or $p < 0.01$) would address these points.

Reviewer #3 (Remarks to the Author):

The revised manuscript by Nath et al. is significantly improved with extended analyses. However, I still have several concerns, as detailed below.

1. Line 150-153:

"As individual genes are variable and sparse in single-cell RNA sequencing data and therefore not commonly shared across patients (Supplementary Figure 2), we used pathway analysis for subsequent analyses."

First, this should refer to Fig. S3 rather than S2. Second, the sentence is phrased in manner that makes it sound as if the lack of consistency between patients in the genes that correlate with time points is merely a technical effect rather than a real biological result; while there is certainly a technical component that makes it difficult to interpret the data, I think that ultimately this data points to an overall observation that progression looks differently in each patient, such that the vast majority of expression changes are patient-specific. with possibly some limited consistent patterns. While this could be conceived as a negative result (i.e. lack of a major consistent pattern), I think that the authors should better acknowledge and clarify this pattern.

2. Line 297-298:

"This relative shift between the archetypes impacted the overall survival of the patients."

This sentence reflects misinterpretation of an observed correlation as indicating a causal effect. It is definitely possible that the archetypes only correlate with, but do not not cause (i.e. "impacted") the survival difference.

3. Lines 299-301:

"While this sample size is small, we found a negative correlation between the proportion of MAP specialists in a patient at the final time point and overall survival (Pearson's correlation = -0.69, R2 = 0.48) (Supplementary Figure 15D)."

Fig. S15D shows that this result is associated with a p-value of 0.1 and hence cannot be considered as statistically significant. While the authors note the small sample size this was not clear from the main text and should be clarified. More importantly, this result is described and even emphasized in the abstract and discussion. Such emphasis of a statistically non-significant result seems problematic to me, especially when the lack of significance is somewhat hidden in a supplementary figure.

4. Abstract (lines 35-37):

"There was a selection for the metabolism and proliferation archetype and against the cellular defense response in cancer cells that received multiple lines of treatment."

The authors refer to an increase in the proportion of a state as "selection" for that state, which implicitly assumes that the mechanism by which the state frequency increased is known and is related to increased fitness. However, there is no support for such hypothesis, and alternative models (e.g. cellular plasticity) cannot be ruled out.

REVIEWER COMMENTS

Reviewer #1 (Remarks to the Author):

The authors adequately addressed my concerns to my satisfaction.

Reviewer #2 (Remarks to the Author):

Reviewer 2: the authors have addressed all of my concerns and my suggestions diligently, except for point 3 of the original rebuttal letter.

The choice of 3 archetypes now looks compelling and is well-supported by Fig S4A.

But I can't say that the authors convinced me that a polytope is a good fit for the data. Showing that a polytope is a good fit for the data statistically-speaking is important because the trade-off interpretation rests on the claim that the data fits a polytope.

Specifically, I have 3 issues with the panel on the t-ratio test in Fig S4B:

1. The authors represent the ratio of the volume of the best-fitting polytope over the volume of the convex hull (CH). Because the CH has more degrees of freedom and because no data point can be outside the polytope, the CH always fits the data better than a polytope. Thus, the volume of the CH is smaller than the volume of the polytope, so that the ratio shown on Fig S4B should be larger than 1. Yet, in Fig S4B, this ratio is always smaller than 1. How can this be?

2. I will assume that there was a labeling mistake in Fig S4B, and that the ratio represented is actually the volume of the CH over the volume of the best-fitting polytope. I will also assume that this ratio in many randomized datasets is represented as a probability density in Fig S4B, together with the t-ratio in the non-randomized data as a vertical line (is this correct? If so, please specify in the legend). There, we see that the t-ratio is increased in randomized data compared to non-randomized data. This implies that a polytope fits the randomized data better than the original, non-randomized data. Such an observation that does not support that a polytope fits the data.

3. Even if I assume that the densities and the line represented in the t-ratio panel of Fig S4B represent other ratios than the one I guessed, the p-values shown do not support the claim that a polytope is a better fit to the data compared to randomized data. The p-value for 3 archetypes is apparently 0.9, which is overwhelming evidence for accepting the null hypothesis (the polytope is a bad fit). A minimal p-value threshold to reject the null hypothesis and thus conclude that the polytope is a good fit would be 0.05. But even this threshold implies too large a type I error so that a threshold of 1% is recommended by statisticians (Ioannidis, PLoS Medicine 2005).

These points need to be addressed showing that a polytope is a statistically significant fit for the data underlies the trade-off interpretation.

Perhaps the easiest way to do so is to use the ParTI software (Hart et al. Nature Methods 2015) with 3 archetypes in order to compute the p-value of t-ratio test. Showing a significant p-value there ($p < .05$ or $p < .01$) would address these points.

We thank the reviewer for their assessment of our revised manuscript and for pointing out the issues with the t-ratio calculation and p-values. We agree the calculation and presentation of t-ratio in the R-package ParetoTI are not well documented. Therefore, we have removed the panel with histogram of t-ratios from Supplementary Figure 4B. As listed by the reviewer, we have now used the MATLAB implementation of the Pareto Task Inference software (ParTI) to confirm the number of archetypes and perform the t-ratio test to obtain a p-value. Our analysis confirmed the selection of 3 archetypes and showed that the t-ratio test has significant p-value below the

threshold recommended by the reviewer ($P < 0.01$). These results support that a 3-vertex polytope is a statistically significant fit for the data. We have updated Supplementary Figure 4 with the result from ParTI analysis of our scRNA-seq data.

The output from the MATLAB ParTI_lite and ParTI analysis are displayed below for reference.

```
>> HGSOC_ParTI
Converting discrete features to booleans
Starting to perform PCA, for big data on slow computers this may take a while...
Calculating explained variance with PCHA (Morup M, Hansen KL, 2011)
Elbow method suggests 3 archetypes.
Please indicate the desired number of archetypes (or press enter for using the suggestion):
Calculating archetypes positions with SISAL (Bioucas-Dias JM, 2009)
finished finding the archetypes
Skipping enrichment analysis as no features were provided.
Converting discrete features to booleans
Starting to perform PCA, for big data on slow computers this may take a while...
Calculating explained variance with PCHA (Morup M, Hansen KL, 2011)
Elbow method suggests 3 archetypes.
Please indicate the desired number of archetypes (or press enter for using the suggestion): 3
Calculating archetypes positions with SISAL (Bioucas-Dias JM, 2009)
finished finding the archetypes
Now computing t-ratios.
10% done
20% done
30% done
40% done
50% done
60% done
70% done
80% done
90% done
100% done
The significance of 3 archetypes has p-value of: 0.00000
Now calculating errors on the archetypes.
10% done
20% done
30% done
40% done
50% done
60% done
70% done
80% done
90% done
100% done
finished finding the archetypes error distribution
Finished sorting data points.
Finished computing discrete enrichments.
Finished computing continuous enrichments.
>>
```

To reflect results from these additional analyses, we have added the following lines to the methods section on Page 36:

“Additionally, we analyzed the CCA-normalized counts data to verify the number of archetypes using the ParTI package for MATLAB (25). We calculated the variance explained by increasing number of archetypes and confirmed that the three archetypes were optimal using the elbow method. A t-ratio test ($P < 0.01$) confirmed that the polytope was a statistically significant fit for the data.”

Reviewer #3 (Remarks to the Author):

The revised manuscript by Nath et al. is significantly improved with extended analyses. However, I still have several concerns, as detailed below.

We thank the reviewer for their careful assessment. Based on their advice, we have tempered the statements in the manuscript as described below.

1. Line 150-153:

"As individual genes are variable and sparse in single-cell RNA sequencing data and therefore not commonly shared across patients (Supplementary Figure 2), we used pathway analysis for subsequent analyses." First, this should refer to Fig. S3 rather than S2. Second, the sentence is phrased in manner that makes it sound as if the lack of consistency between patients in the genes that correlate with time points is merely a technical effect rather than a real biological result; while there is certainly a technical component that makes it difficult to interpret the data, I think that ultimately this data points to an overall observation that progression looks differently in each patient, such that the vast majority of expression changes are patient-specific. with possibly some limited consistent patterns. While this could be conceived as a negative result (i.e. lack of a major consistent pattern), I think that the authors should better acknowledge and clarify this pattern.

We agree with the reviewer's interpretation. The observed lack of consistent pattern of single-gene changes over time was a motivation for us to perform the archetype analysis to project the scRNA-seq data in a low-dimensional space and study the evolution of phenotypes. Accordingly, we have added the following to lines 150-156:

"We analyzed patterns of expression changes over time and found that few differentially expressed genes were commonly shared across patients (Supplementary Figure 3). This pattern could also reflect the sparsity of scRNA-seq data that contributed to the observed lack of consistent changes across patients in the high-dimensional gene expression space. Therefore, we next adopted an approach to project the scRNA-seq data in a low-dimensional space and investigate the evolution of key phenotypes."

2. Line 297-298:

"This relative shift between the archetypes impacted the overall survival of the patients." This sentence reflects misinterpretation of an observed correlation as indicating a causal effect. It is definitely possible that the archetypes only correlate with, but do not not cause (i.e. "impacted") the survival difference.

The sentence has been modified to "This relative shift between the archetypes was correlated with the overall survival of the patients"

3. Lines 299-301:

"While this sample size is small, we found a negative correlation between the proportion of MAP specialists in a patient at the final time point and overall survival (Pearson's correlation = -0.69, $R^2 = 0.48$) (Supplementary Figure 15D)."

Fig. S15D shows that this result is associated with a p-value of 0.1 and hence cannot be considered as statistically significant. While the authors note the small sample size this was not clear from the main text and should be clarified. More importantly, this result is described and even emphasized in the abstract and discussion. Such emphasis of a statistically non-significant result seems problematic to me, especially when the lack of significance is somewhat hidden in a supplementary figure.

We have duly noted this issue and modified lines 301– 303 "This relative shift between the archetypes was correlated with the overall survival of the patients (time to death in days since primary surgery was performed). We found a negative correlation between the proportion of MAP specialists in a patient at the final time point and overall survival (Supplementary Figure 15D). Although this correlation is not statistically significant in this small sample ($P = 0.1$), the observed effect size (Pearson's correlation = -0.69, $R^2 = 0.48$) calls for validation in a sample of larger size to assess clinical significance."

4. Abstract (lines 35-37):

"There was a selection for the metabolism and proliferation archetype and against the cellular defense response in cancer cells that received multiple lines of treatment."

The authors refer to an increase in the proportion of a state as "selection" for that state, which implicitly assumes that the mechanism by which the state frequency increased is known and is related to increased fitness. However, there is no support for such hypothesis, and alternative models (e.g. cellular plasticity) cannot be ruled out.

We understand the confusion arising from the description of the shift in populations as "selection". Therefore, we modified the sentence in the abstract to:

"There was a shift in favor of the metabolism and proliferation archetype versus the cellular defense response archetype in cancer cells that received multiple lines of treatment."

REVIEWERS' COMMENTS

Reviewer #2 (Remarks to the Author):

The authors have addressed all my concerns. I recommend the manuscript for publication.

Reviewer #3 (Remarks to the Author):

The authors have now adequately addressed my concerns.